# AUTOREGRESSIVE DIRECT PREFERENCE OPTIMIZATION

## ABSTRACT

Direct preference optimization (DPO) has emerged as a promising approach for aligning large language models (LLMs) with human preferences. However, the widespread reliance on the response-level Bradley-Terry (BT) model may limit its full potential, as the reference and learnable models are assumed to be autoregressive only after deriving the objective function. Motivated by this limitation, we revisit the theoretical foundations of DPO and propose a novel formulation that explicitly introduces the autoregressive assumption prior to applying the BT model. Specifically, we first reformulate the origin of DPO using two Boltzmann distributions with reward-based energies defined over the output (response) space $\mathcal{Y}$. We then extend the energy domain from $\mathcal{Y}$ to its prefix closure $\mathcal{Y}^*$. Interestingly, this simple extension naturally leads to the energy definitions with an autoregressive reference model, the prefix-wise BT model, and ultimately, a novel DPO variant called **Autoregressive DPO (ADPO)** with its corresponding loss function. Without violating the theoretical foundations, the derived loss takes an elegant form: it shifts the summation operation in the DPO objective outside the log-sigmoid function. Furthermore, through theoretical analysis of ADPO, we show that there exist two length measures to be considered when designing DPO-based algorithms: the token length $\mu$ and the feedback length $\mu'$. To the best of our knowledge, we are the first to explicitly distinguish these two measures and analyze their implications for preference optimization in LLMs.

## 1 INTRODUCTION

Reinforcement learning from human feedback (RLHF) (Christiano et al., 2017; Stiennon et al., 2020; Ouyang et al., 2022a) has emerged as a powerful paradigm for aligning large language models (LLMs) with human preferences. As an alternative to traditional RLHF, Direct Preference Optimization (DPO) (Rafailov et al., 2023) analytically derives the optimal solution to the reward maximization objective, enabling efficient and scalable training. This serves as a foundation for state-of-the-art LLMs, supporting diverse tasks ranging from conversational interactions to complex mathematical reasoning (Zhang et al., 2024b; Pang et al., 2024; Jiao et al., 2025; Liao et al., 2025; Yang et al., 2025b; Shao et al., 2025).

Recent studies have introduced numerous variants of DPO primarily aimed at improving performance of LLMs (Pang et al., 2024; Zhao et al., 2024; Xiao et al., 2024; Meng et al., 2024; Zhang et al., 2024b; Chen et al., 2024; Wu et al., 2024b; Gupta et al., 2025; Lai et al., 2025; Li et al., 2025b; Wu et al., 2025; Son et al., 2025; Hu et al., 2025; Chen et al., 2025; Liang et al., 2025; Yoon et al., 2025). However, most of them, including those proposing token-level objective functions (Zeng et al., 2024; Zhu et al., 2025; Lin et al., 2025), fundamentally rely on the Bradley-Terry (BT) model (Bradley & Terry, 1952), which stipulates the human preference distribution at the level of complete responses. In this modeling, there is an underlying assumption arising from RLHF that the reward function $r(x, y)$ is defined over complete responses $y \in \mathcal{Y}$, each paired with an input $x \in \mathcal{X}$.

This assumption is regarded as nearly essential because reward models typically learn to evaluate each input-output pair $(x, y)$. For LLM training, even though models produce autoregressive distributions $\pi(y_i|y_{<i}, x)$, learning reward models at a finer granularity (*e.g.*, at the token level) is challenging, as it is unrealistic to ask users to evaluate pairs $(x, y_i)$ conditioned on incomplete re-

Table 1: **DPO versus ADPO.** By defining reward-based energy functions over the prefix closure $\mathcal{Y}^*$ of the output space $\mathcal{Y}$, we introduce the prefix-wise Bradley-Terry model. The derived loss shifts the summation operation outside the log-sigmoid function.

| Method | Energy domain | Modeling | Loss |
|---|---|---|---|
| DPO | Output space $\mathcal{Y}$ | Bradley-Terry | $-\log\sigma\left(\beta\sum_i\log\frac{\pi_\theta(y_i^w|y_{<i}^w,x)}{\pi_{\text{ref}}(y_i^w|y_{<i}^w,x)} - \beta\sum_i\log\frac{\pi_\theta(y_i^l|y_{<i}^l,x)}{\pi_{\text{ref}}(y_i^l|y_{<i}^l,x)}\right)$ |
| ADPO (Ours) | Prefix closure $\mathcal{Y}^*$ | Prefix-wise Bradley-Terry | $-\sum_i\log\sigma\left(\beta\log\frac{\pi_\theta(y_i^w|y_{<i}^w,x)}{\pi_{\text{ref}}(y_i^w|y_{<i}^w,x)} - \beta\log\frac{\pi_\theta(y_i^l|y_{<i}^l,x)}{\pi_{\text{ref}}(y_i^l|y_{<i}^l,x)}\right)$ |

sponses $y_{<i}$. However, since DPO does not require explicit reward models, there is an opportunity to introduce a more structured implicit reward function that better aligns with autoregressive modeling.

In this paper, we investigate how the autoregressive assumption can be explicitly incorporated into DPO when applying the BT model. Specifically, we revisit the theoretical foundations of DPO and introduce a novel autoregressive variant, which we call **Autoregressive DPO (ADPO)**. The core idea behind ADPO lies in the introduction of an implicit reward function $r^*(x, y_{<i})$ defined over the prefix closure $\mathcal{Y}^*$, a set of incomplete responses. ADPO is then naturally derived through (1) reward-based prefix energies defined with an autoregressive reference model, (2) their corresponding Boltzmann distributions, and (3) the prefix-wise BT model. The resulting loss function has an elegant form: the summation operation in the DPO loss is shifted outside the log-sigmoid function, as summarized in Table 1. To the best of our knowledge, we are the first to formulate and analyze such an autoregressive extension. It is also worth noting that ADPO does not violate the theoretical foundations of DPO, as the difference naturally arises from the definition of the energy functions. In summary, our contributions are two-fold.

1) We introduce **ADPO**, a novel DPO variant derived by extending the domain of energy from the output space $\mathcal{Y}$ to its prefix closure $\mathcal{Y}^*$. This enables the incorporation of the autoregressive assumption into DPO when applying the BT model, providing a theoretically consistent way to facilitate finer-granularity training.

2) We prove under mild assumptions that any reward function can be reparameterized by **an autoregressive model (Theorem 1)**. Furthermore, through theoretical analysis, we identify two sequence length measures on $\mathcal{Y}$ that are critical for designing DPO-based algorithms: *the token length measure* $\mu$ and *the feedback length measure* $\mu'$. DPO corresponds to the special case of ADPO with $\mu'(y) = 1$ ($\forall y$) (Corollary 1), while setting $\mu' \equiv \mu$ yields a token-level variant. In general, choosing these two measures independently enables training at arbitrary granularity.

## 2 RELATED WORK

**Preference Optimization.** RLHF frames alignment as optimizing a reward based on pairwise human preferences (Christiano et al., 2017; Stiennon et al., 2020; Ouyang et al., 2022a; Xiong et al., 2024; Ye et al., 2024; Zhu et al., 2023). Online alignment methods maximize expected return under on-policy rollouts by employing Kullback–Leibler (KL) regularization techniques, such as trust-region constraints or explicit penalties relative to a reference policy (Schulman et al., 2015; 2017; Ouyang et al., 2022b). For LLM training, PPO-based methods have explored improved trade-offs between return and divergence (Wu et al., 2023; Zhang et al., 2024a; Wu et al., 2024a; Dai et al., 2024; Zhang et al., 2025), leading to extended approaches such as GRPO (Shao et al., 2024a). For efficient training, DPO (Rafailov et al., 2023) has emerged as a primary choice because it significantly reduces computational overhead by eliminating the need for an explicit reward model. Within these methods, foundational probabilistic models for pairwise choice, such as the Bradley-Terry and Plackett-Luce models (Bradley & Terry, 1952; Plackett, 1975), remain standard for preference modeling and are typically applied at the level of complete responses.

**Granularity of DPO.** Several recent studies have proposed granularity-aware methods to better align with autoregressive generation. For example, SimPO employs a length-normalized loss function (Meng et al., 2024), TDPO imposes forward-KL constraints at each token position (Zeng et al., 2024), TGDPO introduces token-level reward guidance (Zhu et al., 2025), and cDPO (Lin et al., 2025) identifies critical tokens to apply token-level weights. Despite these advances, existing methods often fundamentally rely on the vanilla BT model at the level of complete responses. In contrast, our formulation integrates the autoregressive assumption into DPO when applying the BT model.

Beyond these granularity-oriented approaches, another line of work revisits DPO from the perspective of reinforcement learning. In particular, Rafailov et al. (2024) provides a complementary analysis showing that standard DPO can be interpreted at the token level through a soft-Q-learning lens, where the model's logits correspond to an optimal Q-function and the implicit reward aligns with an optimal advantage function. While this analysis explains the implicit token-level structure of the original response-level BT formulation, it does not modify the BT model itself. In contrast, our formulation incorporates the autoregressive assumption before applying the BT model by extending the energy domain to the prefix closure $Y^*$, yielding a prefix-wise BT model and a distinct objective.

## 3 PRELIMINARY

DPO Rafailov et al. (2023) is a preference optimization algorithm that directly optimizes models using human preferences without relying on an explicit reward model. To simplify the derivation and interpretation of its objective function, we reformulate DPO using two Boltzmann distributions. We begin by defining the two reward-based energy functions, which we refer to as the DPO energies.

***Definition 1 (DPO Energies).*** *Given a reward function $r(x, y)$ and a reference model $\pi_{\text{ref}}(y|x)$, we define the two DPO energies, namely the likelihood energy $E_1$ and the posterior energy $E_2$:*

$$E_1(x, y) = -r(x, y), \quad (1) \qquad E_2(x, y) = -\frac{1}{\beta} r(x, y) - \log \pi_{\text{ref}}(y|x), \quad (2)$$

*where $x \in \mathcal{X}$ is an input from the input space $\mathcal{X}$, $y \in \mathcal{Y}$ is the model's output (response) in the output space $\mathcal{Y}$, and $\beta \in \mathbb{R}_+$ is a hyperparameter.*

These energy functions derive the two corresponding Boltzmann distributions:

$$p_1(y^w \succ y^l | x) = \frac{\exp(-E_1(x, y^w))}{\sum_{y \in Y} \exp(-E_1(x, y))}, \quad (3) \qquad p_2(y|x) = \frac{\exp(-E_2(x, y))}{\sum_{y \in \mathcal{Y}} \exp(-E_2(x, y))}, \quad (4)$$

where $p_1$ is normalized over a set of preferred and dispreferred responses $Y = \{y^w, y^l\} \subset \mathcal{Y}$ following the BT model (Bradley & Terry, 1952), whereas $p_2$ is normalized over the entire output space $\mathcal{Y}$. It is straightforward to see that $p_2$ maximizes the KL-constrained reward function (Peters & Schaal, 2007; Peng et al., 2019; Korbak et al., 2022; Go et al., 2023; Rafailov et al., 2023). Then, DPO minimizes the negative log-likelihood loss $\mathcal{L}$ with respect to $p_1$:

$$\mathcal{L}_{\text{DPO}} = -\mathbb{E}_{(x,Y)\sim\mathcal{D}} \big[ \log p_1(y^w \succ y^l | x) \big] = -\mathbb{E}_{(x,Y)\sim\mathcal{D}} \left[ \log \sigma \left( \beta \log \frac{p_2(y^w|x)}{\pi_{\text{ref}}(y^w|x)} - \beta \log \frac{p_2(y^l|x)}{\pi_{\text{ref}}(y^l|x)} \right) \right]$$
(5)

where $\sigma$ is the sigmoid function and $\mathcal{D}$ is the preference dataset. During training, $p_2$ is parameterized as $p_2 = \pi_\theta$ with a parameter set $\theta$. This is the reparameterization trick of DPO.

In the recent paradigm of LLMs, models are typically assumed to be autoregressive, *i.e.*, assuming that $\mathcal{Y}$ is a space of sequence of arbitrary length, both reference and learnable models satisfy the chain rule: $\pi(y|x) = \prod_{i=1}^{T} \pi(y_i | y_{<i}, x)$, where $T = \mu(y)$ is the length of $y$.[1] From this, we have

$$\mathcal{L}_{\text{DPO}} = -\mathbb{E}_{(x,Y)\sim\mathcal{D}} \left[ \log \sigma \left( \sum_{i=1}^{T} \left( \beta \log \frac{\pi_\theta(y_i^w | y_{<i}^w, x)}{\pi_{\text{ref}}(y_i^w | y_{<i}^w, x)} - \beta \log \frac{\pi_\theta(y_i^l | y_{<i}^l, x)}{\pi_{\text{ref}}(y_i^l | y_{<i}^l, x)} \right) \right) \right], \quad (6)$$

where right-padding is applied so that both sequences have the same length following practical implementation of DPO using mini-batch optimization algorithms.

## 4 AUTOREGRESSIVE DIRECT PREFERENCE OPTIMIZATION

From the derivation of DPO, we observe an important discrepancy: while the learnable model $\pi_\theta$ is autoregressive, the Boltzmann distribution $p_2$ defined in Eq. (4) is not formulated as such. This motivates our research question: *Can we define energy functions that yield $p_2$ explicitly as autoregressive distribution?* Our ADPO formulation answers this question affirmatively.

---

[1] $\mu : \mathcal{Y} \to \mathbb{N}$ is the token length measure that assigns to each $y \in \mathcal{Y}$ the number of tokens it contains.

The core idea behind ADPO lies in the introduction of the ADPO energies, which define energies and the corresponding implicit reward function over the prefix closure of the output space.

**Definition 2 (ADPO Energies).** *Let $\mathcal{Y}^*$ be the prefix closure of $\mathcal{Y}$, i.e.,*

$$\mathcal{Y}^* = \bigcup_{y \in \mathcal{Y}} \{y_{\leq i} = (y_1, y_2, \cdots, y_i) : 0 \leq i \leq T'\}, \tag{7}$$

*where $T' = \mu'(y)$ is the length of $y$.[2] Given a __prefix-wise__ reward function $r^* : \mathcal{X} \times \mathcal{Y}^* \to \mathbb{R}$ and an __autoregressive__ reference model $\pi_{\text{ref}}(y|x)$, we define the two ADPO energies, namely prefix likelihood energy $E_1^*$ and the prefix posterior energy $E_2^*$, as follows:*

$$E_1^*(x, y_{\leq i}) = -r^*(x, y_{\leq i}), \qquad (8) \qquad E_2^*(x, y_{\leq i}) = -\frac{1}{\beta} r^*(x, y_{\leq i}) - \log \pi_{\text{ref}}(y_i|y_{<i}, x). \tag{9}$$

There are two main differences compared to Definition 1, as underlined in Definition 2. First, the reward function is defined over $\mathcal{X} \times \mathcal{Y}^*$ instead of $\mathcal{X} \times \mathcal{Y}$. With this, we define the prefix-wise BT model by taking product of BT-based preference distributions as

$$p_1(y^w \succ y^l|x) = \prod_{i=1}^{T'} \frac{\exp(-E_1^*(x, y_{\leq i}^w))}{\sum_{y_{\leq i} \in Y_i} \exp(-E_1^*(x, y_{\leq i}))}, \tag{10}$$

where $Y_i = \{y_{\leq i}^w, y_{\leq i}^l\} \subset \mathcal{Y}^*$ is a set of preferred and dispreferred responses up to length $i$. Second, the reference model is assumed to be autoregressive at this stage (in Eq. (9)). With this, the second distribution $p_2$ also becomes autoregressive prior to reparameterization:

$$p_2(y|x) = \prod_{i=1}^{T'} p_2(y_i|y_{<i}, x), \quad p_2(y_i|y_{<i}, x) = \frac{\exp(-E_2^*(x, y_{\leq i}))}{\sum_{y_i \in \mathcal{V}} \exp(-E_2^*(x, y_{\leq i}))}, \tag{11}$$

where $\mathcal{V} = \{y \in \mathcal{Y} : \mu'(y) = 1\}$. This alleviates the mismatch between $p_2$ and the learnable autoregressive model $\pi_\theta$.

**ADPO Objective.** Analogous to DPO, ADPO maximizes the log-probability of $p_1(y^w \succ y^l|x)$ during training. Specifically, we define the loss function as

$$\mathcal{L}_{\text{ADPO}} = -\mathbb{E}_{(x,Y) \sim \mathcal{D}} \left[ \log p_1(y^w \succ y^l|x) \right]. \tag{12}$$

Through reparameterization, this loss function can be written as

$$\mathcal{L}_{\text{ADPO}} = -\mathbb{E}_{(x,Y) \sim \mathcal{D}} \left[ \sum_{i=1}^{T'} \log \sigma \left( \beta \log \frac{\pi_\theta(y_i^w|y_{<i}^w, x)}{\pi_{\text{ref}}(y_i^w|y_{<i}^w, x)} - \beta \log \frac{\pi_\theta(y_i^l|y_{<i}^l, x)}{\pi_{\text{ref}}(y_i^l|y_{<i}^l, x)} \right) \right]. \tag{13}$$

A proof is provided in Appendix A. Interestingly, ADPO shifts the summation operation in the DPO loss in Eq. (6) outside the log-sigmoid function. In terms of KL-constrained reward maximization, the solution of ADPO remains the optimal solution as shown in Appendix B.

## 5 THEORETICAL ANALYSIS

ADPO does not violate the theoretical foundations of DPO. Rather, the theory is generalized by bridging additive reward functions and autoregressive models. Ultimately, we prove that, under mild assumptions, any reward function can be reparameterized by an autoregressive model (**Theorem 1**). This better aligns the DPO theory with the paradigm of autoregressive LLMs.

### 5.1 REPARAMETERIZATION COMPLETENESS

We first show how prefix-wise reward functions can be reparameterized by autoregressive models.

**Proposition 1 (Prefix-wise Reparameterization Completeness).** *Let $[r^*]$ denote the reward-shift equivalence class of a prefix-wise reward function $r^* : \mathcal{X} \times \mathcal{Y}^* \to \mathbb{R}$, defined as*

$$[r^*] = \{r' \mid \exists f \; \forall(x, y_{\leq i}) \; r'(x, y_{\leq i}) = r^*(x, y_{\leq i}) + f(x, y_{<i})\}. \tag{14}$$

---

[2]We will discuss later that this length measure can be independent of the token length measure $\mu$ but one can assume $\mu' = \mu$ to improve readability.

*Given an autoregressive reference model $\pi_{\text{ref}}(y_i|y_{<i}, x) > 0$ and a hyperparameter $\beta > 0$, for any prefix-wise reward function $r^*$, there exists a unique representative $r^*_\circ \in [r^*]$ such that, for all $x \in \mathcal{X}$ and prefixes $y_{\leq i} \in \mathcal{Y}^*$,*

$$r^*_\circ(x, y_{\leq i}) \equiv \beta \log \frac{\pi(y_i|y_{<i}, x)}{\pi_{\text{ref}}(y_i|y_{<i}, x)}. \tag{15}$$

*for some autoregressive model $\pi$.*

A proof is provided in Appendix C. To bridge vanilla reward functions $r(x, y)$ and prefix-wise reward functions $r^*(x, y_{\leq i})$, we define the additive decomposition of $r$. Obviously, Lemma 1 holds.

**Definition 3 (Additive Decomposition).** *Let $r : \mathcal{X} \times \mathcal{Y} \to \mathbb{R}$ be a reward function. We say that a prefix-wise reward function $r^* : \mathcal{X} \times \mathcal{Y}^* \to \mathbb{R}$ is an additive decomposition of $r$ if and only if $r(x, y) = \sum_{i=1}^{T'} r^*(x, y_{\leq i})$ for any $x \in \mathcal{X}$ and $y \in \mathcal{Y}$.*

**Lemma 1.** *Every reward function has an additive decomposition.*

Then, from Proposition 1 and Lemma 1, it follows that Theorem 1 holds.

**Theorem 1.** *All reward classes consistent with the prefix-wise Bradley-Terry models can be represented with the reparameterization $r(x, y) = \beta \log \frac{\pi(y|x)}{\pi_{\text{ref}}(y|x)}$ for some autoregressive model $\pi(y|x)$ and a given autoregressive reference model $\pi_{\text{ref}}(y|x)$ such that $\pi(y_i|y_{<i}, x) > 0$.*

*Proof.* For any reward function $r$, its additive decomposition $r^*$ exists from Lemma 1. From Proposition 1, there exists an autoregressive model $\pi$ such that

$$r(x, y) = \sum_{i=1}^{T'} r^*(x, y_{\leq i}) \equiv \sum_{i=1}^{T'} r^*_\circ(x, y_{\leq i}) = \sum_{i=1}^{T'} \beta \log \frac{\pi(y_i|y_{<i}, x)}{\pi_{\text{ref}}(y_i|y_{<i}, x)} = \beta \log \frac{\pi(y|x)}{\pi_{\text{ref}}(y|x)}, \tag{16}$$

where $\equiv$ denotes equality in the equivalence class. $\square$

In contrast to the DPO's theorem (Rafailov et al., 2023), Theorem 1 explicitly demonstrates reparameterizability using an autoregressive model.

## 5.2 Implicit length measure in DPO

The following corollary further bridges DPO and ADPO, providing deeper insights into DPO.

**Corollary 1.** *When the length $\mu'(y)$ is one for all $y \in \mathcal{Y}$, ADPO reduces to DPO that relies on the traditional (non-prefixwise) Bradley-Terry model.*

This result is intuitively difficult to grasp. One might straightforwardly interpret "length equals one for all $y$" as restricting the output space $\mathcal{Y}$ to a single-token space, implying that ADPO coincides with DPO in this special case. While this interpretation is correct, it does not fully capture the true meaning of the corollary. Rather, our interpretation emphasizes that the output space $\mathcal{Y}$ remains a sequence space. Then, Corollary 1 shows that, underlying the original DPO formulation, there exists **an implicit length measure $\mu' : \mathcal{Y} \to \mathbb{N}$, which assigns a length of one $\mu'(y) = 1$ to every sequence** $y \in \mathcal{Y}$.

This measure may have limited the exploration of research ideas for DPO-based formulations in previous studies, as it predisposed most existing methods to place the summation operation inside the log-sigmoid function, as in Eq. (6). We further discuss its deeper roots.

## 5.3 Two Distinct Length Measures

**What is $\mu'$?** Although the implicit length measure $\mu'$ may appear counterintuitive, it is natural to interpret this measure as arising from RLHF. Specifically, $\mu'$ measures the length of sequences $y$ in an evaluation scenario by first mapping sequences into a one-dimensional space using an evaluation metric $\nu : \mathcal{Y} \to \mathbb{R}$, subsequently defining length as $\mu'(y) = \text{length}(\nu(y)) = \dim(\mathbb{R}) = 1$. This interpretation is particularly natural when human feedback is performed on complete responses. Therefore, we refer to $\mu'$ as the *feedback length measure*.

**Two Length Measures are Independent.** The discussion thus far suggests two algorithms, as summarized in Table 2. The first is DPO, corresponding to the case where $\mu' \equiv 1$, which evaluates

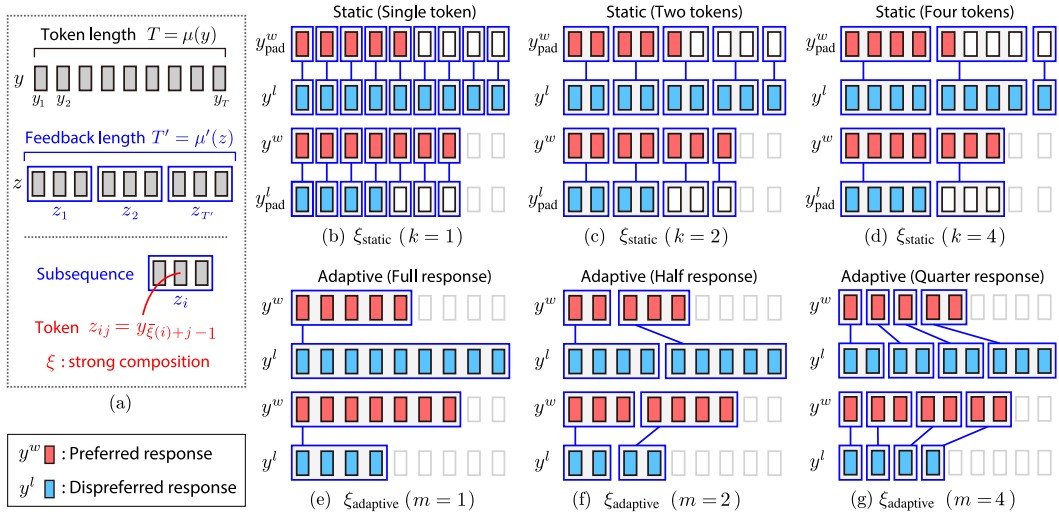

Figure 1: Static and adaptive families of ADPO. (a) Token and feedback length measures. Each subsequence $z_i$ is defined with strong composition $\xi$. (b-d) Static families. Each response is decomposed into $T/k$ subsequences depending on the token length $T$ with a fixed window size $k$. (e-g) Adaptive families. Each response is uniformly decomposed into $m$ subsequences. (e) is equivalent to DPO. Regions enclosed by blue rectangles indicate segments over which summation operation is applied within the log-sigmoid function of the loss function.

the complete response as a whole; thus, we denote its granularity as "Full Response" in the table. The second is token-level ADPO, corresponding to $\mu' = \mu$, where $\mu$ is the token length measure induced by LLMs. This evaluates each token individually, and its granularity is indicated as "Single Token".

Table 2: Granularity

| Method | Granularity | $\mu'$ |
|--------|-------------|--------|
| DPO | Full Response | 1 |
| ADPO | Single Token | $\mu$ |

However, importantly, ADPO itself does not impose the constraint $\mu' = \mu$. Although the two length measures should be related to each other in practice, these measures can theoretically be independent because the origins of the measures, one from LLMs and the other from a prefix closure, are independent. This leads to granularity families of ADPO, which we discuss in the next subsection.

## 5.4 GRANULARITY FAMILIES

ADPO allows any intermediate granularity with $1 \leq \mu'(y) \leq \mu(y)$ in practice. Specifically, by decomposing each sequence $y$ into subsequences $\{z_i\}_{i=1}^{T'}$ such that each prefix $z_{\leq i} = (z_1, z_2, \ldots, z_i)$ constitutes a unit eligible for (implicit) feedback, the ADPO loss can be written as follows:

$$\mathcal{L}_{\text{ADPO}} = -\mathbb{E}_{(x,Y)\sim\mathcal{D}}\left[\sum_{i=1}^{T'} \log \sigma\left(\beta\sum_{j=1}^{T_i^w} \log \frac{\pi_\theta(z_{i,j}^w|z_{<i,<j}^w,x)}{\pi_{\text{ref}}(z_{i,j}^w|z_{<i,<j}^w,x)} - \beta\sum_{j=1}^{T_i^l} \log \frac{\pi_\theta(z_{i,j}^l|z_{<i,<j}^l,x)}{\pi_{\text{ref}}(z_{i,j}^l|z_{<i,<j}^l,x)}\right)\right],$$
(17)

where $z = (z_1, z_2, \cdots, z_{T'})$ is a decomposition of $y$, $z_i = (z_{i,1}, z_{i,2}, \cdots, z_{i,T_i})$ is a subsequence[3], $T'$ is the feedback length, and $T_i = \mu(z_i)$ is the token length of $z_i$.

More formally, the mapping from $y$ to $z$ is stipulated by a strong composition $\xi$. Specifically, given a token sequence $y = (y_1, \cdots, y_T) \in \mathcal{Y}$, we define each subsequence as

$$z_i = (y_{\bar{\xi}(i)}, y_{\bar{\xi}(i)+1}, \cdots, y_{\bar{\xi}(i+1)-1}), \quad \bar{\xi}(1) = 1, \quad \bar{\xi}(i+1) = \bar{\xi}(i) + \xi(i),$$
(18)

where $\xi \in \mathcal{C}_{T'}(T)$ and $\mathcal{C}_m(n) = \{\xi : [m] \to \mathbb{N}_+ \mid \sum_{j=1}^m \xi(j) = n\}$ is the set of strong compositions of $n \in \mathbb{N}_+$ into exactly $m \in \mathbb{N}_+$ parts, with each part a positive integer.[4] Each token is

---

[3]Each prefix is now denoted by $z_{<i,<j} = (z_1, z_2, \cdots, z_{i-1}, (z_{i,1}, z_{i,2}, \cdots, z_{i,j-1}))$.

[4]$[m] = \{1, 2, \cdots, m\}$

then given by $z_{ij} = y_{\bar{\xi}(i)+j-1}$, as illustrated in Figure 1(a). With this formulation, the choice of $\xi$ determines the learning behavior of ADPO. We define two families of ADPO for practical use.

**Static Family.** This family fixes the window size to $k \in \mathbb{N}_+$ and determines the feedback length depending on the token length. Since the preferred and dispreferred responses may have different token lengths, we first apply EOS padding to equalize lengths, obtaining token sequences $y_{\text{pad}}^w$ and $y_{\text{pad}}^l$ (only the shorter one is padded). Each sequence is then decomposed into $\lceil T/k \rceil$ subsequences, where $T = \mu(y_{\text{pad}}^w) = \mu(y_{\text{pad}}^l)$ is the token length. The strong composition for this family is defined as

$$\xi_{\text{static}}(i) = \min(k, \mu(y_{\text{pad}}^w) - k(i-1)). \tag{19}$$

As illustrated in Figure 1 (b-d), $k = 1$ corresponds to the finest granularity at the token level, while $k > 1$ decreases granularity. The blue rectangles in the figure indicate segments over which summation operation is applied within the log-sigmoid function in Eq. (17).

**Adaptive Family.** This family decomposes each sequence into $m \in \mathbb{N}_+$ subsequences as uniformly as possible and fixes the feedback length as $\mu'(y) = m$ for all sequences $y \in \mathcal{Y}$. The strong composition for this family is defined as

$$\xi_{\text{adaptive}}(i) = \left\lfloor \frac{i\mu(y)}{m} \right\rfloor - \left\lfloor \frac{(i-1)\mu(y)}{m} \right\rfloor. \tag{20}$$

This reduces to DPO when $m = 1$, as illustrated in Figure 1(e). When $m > 2$, both preferred and dispreferred responses are equally partitioned as shown in Figures 1(f) and (g). This naturally extends DPO to finer granularity.

# 6 EXPERIMENTS

## 6.1 EXPERIMENTAL SETUP

**Datasets and Metrics.** We use two representative math reasoning datasets: GSM8K (Cobbe et al., 2021) and MATH500 (Hendrycks et al., 2021). GSM8K is a dataset of grade school math word problems that requires multi-step reasoning to arrive at the correct numerical answer. MATH comprises challenging competition mathematics problems spanning various topics including algebra, geometry, probability, and number theory. All models are fine-tuned on the GSM8K training set. We report accuracy using the 8-shot evaluation protocol for GSM8K (Wei et al., 2022) and the 4-shot evaluation protocol for MATH500 (Lewkowycz et al., 2022).

**Baselines.** We include two baselines: DPO (Rafailov et al., 2023) and cDPO (Lin et al., 2025). cDPO is one of the latest preference optimization methods, which leverages token-level contrastive estimation to identify and penalize critical tokens in incorrect reasoning trajectories.

**ADPO Variants.** We apply our approach to both DPO and cDPO, and refer to the resulting variants as ADPO and cADPO, respectively. Specifically, ADPO uses the loss in Eq. (17) and cADPO extends it to utilize the critical tokens as follows:

$$\mathcal{L}_{\text{cADPO}} = -\mathbb{E}_{(x,Y)\sim\mathcal{D}} \left[ \sum_{i=1}^{T'} \log \sigma \left( \beta \sum_{j=1}^{T_i^w} \log \frac{\pi_\theta(z_{i,j}^w | z_{<i,<j}^w, x)}{\pi_{\text{ref}}(z_{i,j}^w | z_{<i,<j}^w, x)} - \beta \sum_{j=1}^{T_i^l} \bar{s}_j \log \frac{\pi_\theta(z_{i,j}^l | z_{<i,<j}^l, x)}{\pi_{\text{ref}}(z_{i,j}^l | z_{<i,<j}^l, x)} \right) \right], \tag{21}$$

where the weight $\bar{s}_j = 1 - s_j$ is computed based on the token-level reward score $s_j$ obtained from the contrastive estimation method (Lin et al., 2025). We use ADPO and cADPO in the adaptive family with $m = 256$ as our primary methods.

**Models.** We conduct experiments with four diverse LLMs: Llama3-8B-Base (Grattafiori et al., 2024), Qwen-3-8B-Base (Yang et al., 2025a), Gemma-3-12B-PT (Gemma Team, 2025) and DeepSeek-Math-7B (Shao et al., 2024b).

**Implementation Details.** Each model is trained with LoRA using the AdamW optimizer for 3 epochs. Since cDPO requires a larger learning rate, we set it to $2.0 \times 10^{-5}$ for DPO and ADPO, and $4.0 \times 10^{-5}$ for cDPO and cADPO.

Table 3: Experimental results on two mathematical reasoning benchmarks.

| Method | Llama-3-8B | | Gemma-3-12B | | Qwen-3-8B | | DeepSeek-Math-7B | |
| --- | --- | --- | --- | --- | --- | --- | --- | --- |
| | GSM8K | MATH | GSM8K | MATH | GSM8K | MATH | GSM8K | MATH |
| DPO (Rafailov et al., 2023) | 64.37 | 18.00 | 77.03 | 39.80 | 86.96 | 53.80 | 67.78 | 32.00 |
| ADPO (Ours) | **68.08** | **21.00** | **78.32** | **41.20** | **88.10** | **55.40** | **69.98** | **33.40** |
| cDPO (Lin et al., 2025) | 67.90 | 16.80 | 77.18 | 38.60 | 90.98 | 56.80 | 72.90 | 33.40 |
| cADPO (Ours) | **68.76** | **20.20** | **78.85** | **40.40** | **91.74** | **57.20** | **73.54** | **35.40** |

## 6.2 EXPERIMENTAL RESULTS

**Main Results.** Table 3 summarizes the experimental results. We observe that our approach consistently outperforms both the DPO and cDPO baselines across all four LLMs, achieving the best results with cADPO. The highest individual result was obtained by Qwen-3-8B using cADPO, reaching 91.74% on GSM8K. These results demonstrate the practical effectiveness of our approach.

**Granularity Families.** To analyze how varying granularity affects ADPO performance, we systematically explore both static and adaptive families across a range of hyperparameter values in Table 4. For the static family, we observe that models with $k \leq 4$ consistently outperform DPO, indicating the benefit of finer granularity. With DeepSeek-Math-7B, the finest granularity (token-level, $k = 1$) achieves the best performance. However, there remains room for improvement, as the static family requires padding when computing the loss, which potentially constrains its performance.

For the adaptive family, which is equivalent to DPO when $m = 1$, we observe performance degradation across several LLMs when $m$ is small (*e.g.*, $m = 2$). This may occur because dividing responses into a small number of segments can result in unnatural or arbitrary splits. This limitation is addressed by increasing the number of segments. Specifically, when $m \geq 128$, ADPO consistently outperforms DPO in all cases, again indicating the benefit of finer granularity.

When comparing the best-performing models from the static and adaptive families, the adaptive family tends to achieve superior performance. As the adaptive family naturally extends DPO without requiring padding, it effectively addresses the limitations associated with the static family. Overall, these findings validate our theoretical insight that defining the token-length measure and feedback-length measure through strong compositions is effective.

**Learning Behavior.** Figure 2 analyzes the training dynamics of ADPO compared to DPO. Specifically, it shows the evolution of log probabilities for preferred (chosen) and dispreferred (rejected) sequences across training steps for each LLM. We observe that ADPO effectively amplifies the distinction between preferred and dispreferred responses by consistently elevating the probabilities of preferred responses while suppressing those of dispreferred responses. This tendency becomes more pronounced as granularity increases; that is, as $k$ decreases and $m$ increases. These observations underscore the effectiveness of incorporating the autoregressive assumption into the BT model, enabling ADPO variants to more effectively guide models toward preferred outputs across training.

**Conversation Tasks.** To further validate the generalizability of our approach, we conduct experiments on conversation tasks. Specifically, following (Meng et al., 2024), we adopt three widely recognized benchmarks: AlpacaEval 2 (Dubois et al., 2024), Arena-Hard (Li et al., 2025a), and MT-Bench (Zheng et al., 2023). For the training data, we follow the same data construction procedure as SimPO (Meng et al., 2024). Specifically, we generate multiple candidate responses from the base model and determine the chosen/rejected pairs using PairRM annotations (Jiang et al., 2023). We fine-tune Llama-3-8B-Instruct on this dataset and compare SFT, DPO (Rafailov et al., 2023), SimPO, and ADPO. Table 5 summarizes the results. Consistently, ADPO achieves superior performance against DPO across all three benchmarks and performs comparably to or even better than SimPO, underscoring its enhanced capability in generating more preferred responses.

In Table 6, we also find that granularity exhibits a similar trend as in mathematical reasoning tasks: finer-grained ADPO variants consistently outperform coarser granularity configurations in both static and adaptive families. Collectively, these results demonstrate that ADPO not only improves mathematical reasoning but also effectively generalizes across diverse conversational tasks.

Table 4: Granularity families of ADPO. Best results within the static and adaptive families are underlined, and the overall best results are highlighted in bold.

| Method | $\mu'(y)$ | Composition | Granularity | Llama-3-8B GSM | MATH | Gemma-3-12B GSM | MATH | Qwen-3-8B GSM | MATH | DS-Math-7B GSM | MATH |
|--------|-----------|-------------|-------------|------|------|------|------|------|------|------|------|
| ADPO | $\mu(y)/8$ | $\xi_{\text{static}}\ (k\!=\!8)$ | Eight tokens | 64.97 | 18.20 | 76.88 | 40.00 | 87.79 | 54.20 | 68.39 | 32.00 |
| ADPO | $\mu(y)/4$ | $\xi_{\text{static}}\ (k\!=\!4)$ | Four tokens | 66.26 | 17.40 | 77.41 | 40.80 | 88.48 | 54.20 | 68.23 | 32.00 |
| ADPO | $\mu(y)/2$ | $\xi_{\text{static}}\ (k\!=\!2)$ | Two tokens | 66.57 | 18.60 | 77.63 | 41.20 | 88.25 | 54.20 | 70.05 | 33.60 |
| ADPO | $\mu(y)$ | $\xi_{\text{static}}\ (k\!=\!1)$ | Single token | 66.41 | 18.00 | 78.09 | 39.80 | 88.48 | 54.40 | **70.36** | **34.80** |
| DPO | 1 | $\xi_{\text{adaptive}}(m\!=\!1)$ | Full response | 64.37 | 18.00 | 77.03 | 39.80 | 86.96 | 53.80 | 67.78 | 32.00 |
| ADPO | 2 | $\xi_{\text{adaptive}}(m\!=\!2)$ | Half response | 64.29 | 19.60 | 76.88 | 40.20 | 87.95 | 53.80 | 68.61 | 33.40 |
| ADPO | 16 | $\xi_{\text{adaptive}}(m\!=\!16)$ | 1/16 response | 64.52 | 18.00 | 77.10 | 40.80 | 87.26 | 52.40 | 68.84 | 33.40 |
| ADPO | 128 | $\xi_{\text{adaptive}}(m\!=\!128)$ | 1/128 response | 66.72 | 18.20 | 77.63 | 41.00 | 89.16 | 54.60 | 69.83 | 33.60 |
| ADPO | 256 | $\xi_{\text{adaptive}}(m\!=\!256)$ | 1/256 response | **68.08** | **21.00** | **78.32** | **41.20** | 88.10 | **55.40** | 69.98 | 33.40 |
| ADPO | 512 | $\xi_{\text{adaptive}}(m\!=\!512)$ | 1/512 response | 67.32 | 19.00 | 77.94 | 40.00 | 88.02 | 54.00 | **70.36** | 34.40 |

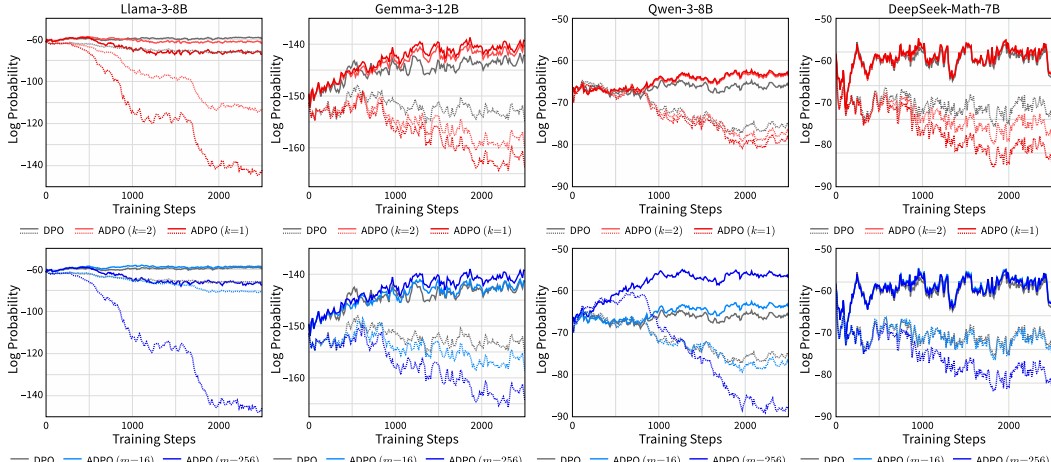

Figure 2: Comparison of training dynamics between DPO and ADPO. The evolution of log probabilities for preferred sequences (solid line) and dispreferred sequences (dashed line) during training is shown. Top row: static family with $k = 1, 2$. Bottom row: adaptive family with $m = 16, 256$.

Table 5: Experimental results on AlpacaEval 2, Arena-Hard, and MT-Bench. Following the official evaluation protocol of each benchmark, length-controlled win rate (LC), win rate (WR), and benchmark scores are reported.

| Method | Llama3-8B AlpacaEval 2 LC | WR | Arena-Hard WR | MT-Bench Score |
|--------|------|------|------|------|
| SFT | 26.0 | 25.3 | 22.3 | 6.9 |
| DPO Rafailov et al. (2023) | 40.3 | 37.9 | 32.6 | 7.0 |
| SimPO Meng et al. (2024) | 44.7 | 40.5 | 33.8 | 7.0 |
| ADPO (Ours) | **45.8** | **41.1** | **34.4** | **7.1** |

Table 6: Granularity families (AlpacaEval 2, Llama3-8B).

| Composition | LC | WR |
|-------------|------|------|
| $\xi_{\text{static}}\ (k\!=\!4)$ | 44.9 | 40.9 |
| $\xi_{\text{static}}\ (k\!=\!2)$ | 45.1 | 40.8 |
| $\xi_{\text{static}}\ (k\!=\!1)$ | **45.8** | 40.8 |
| $\xi_{\text{adaptive}}\ (m\!=\!1)$ | 40.3 | 37.9 |
| $\xi_{\text{adaptive}}\ (m\!=\!2)$ | 41.0 | 40.2 |
| $\xi_{\text{adaptive}}\ (m\!=\!16)$ | 44.1 | 40.2 |
| $\xi_{\text{adaptive}}\ (m\!=\!128)$ | **45.8** | **41.1** |
| $\xi_{\text{adaptive}}\ (m\!=\!256)$ | 44.6 | 40.5 |
| $\xi_{\text{adaptive}}\ (m\!=\!512)$ | 44.7 | 40.3 |

## 7 CONCLUSION

We introduced ADPO, a novel approach for direct preference optimization that integrates the autoregressive assumption when applying the Bradley-Terry model. By forming the prefix closure of the output space, we naturally derived the objective function of ADPO that shifts the summation operation outside the log-sigmoid function. Moreover, we provided a theoretical analysis of ADPO, which

yields deeper insights into the original DPO framework, identifying two distinct length measures. Our experimental evaluations extensively validated ADPO across five benchmarks.

**Limitations and Future Research Directions.** While our study offered novel theoretical and practical perspectives on DPO, it still relies on the original KL-constrained reward maximization problem. Extending beyond this constraint, future research could investigate more general divergence metrics as alternatives to KL divergence, potentially offering greater flexibility and robustness. Moreover, our formulation of the prefix posterior energy in Eq. (9) is specifically motivated by the autoregressive assumption characteristic of LLMs. It would be valuable to explore alternative definitions of energy that are appropriate for other classes of generative models, particularly those that do not adhere strictly to autoregressive architectures. Such investigations could further expand the applicability and effectiveness of DPO-based optimization methods across diverse model types.

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

## A  DERIVATION OF ADPO LOSS

We provide a proof of Eq. (13). We first provide detailed definition of ADPO loss and then show the Proposition A.1.

**Definition A.1.** *We define ADPO loss as*

$$\mathcal{L}_{ADPO} = -\mathbb{E}_{(x,Y) \sim \mathcal{D}} \left[ \log p_1(y^w \succ y^l | x) \right]. \tag{22}$$

*with ADPO energies given by*

$$E_1^*(x, y_{\leq i}) = -r^\circ(x, y_{\leq i}), \tag{23}$$

$$E_2^*(x, y_{\leq i}) = -\frac{1}{\beta} r^\circ(x, y_{\leq i}) - \log \pi_{\text{ref}}(y_i | y_{<i}, x), \tag{24}$$

*where $r^\circ \in [r^*]$ is a reward equivalent to $r^*$ given by*

$$r^\circ(x, y_{\leq i}) = r^*(x, y_{\leq i}) - f(x, y_{<i}), \tag{25}$$

$$f(x, y_{<i}) = \beta \log \sum_{y_i \in \mathcal{V}} \pi_{ref}(y_i | y_{<i}, x) \exp\left( \frac{1}{\beta} r^*(x, y_{\leq i}) \right). \tag{26}$$

**Proposition A.1.** *Given an autoregressive reference model $\pi_{\text{ref}}$, an autoregressive learnable model $\pi_\theta$, a dataset $\mathcal{D}$, and a constant $\beta > 0$, the ADPO loss is given by*

$$\mathcal{L}_{ADPO} = -\mathbb{E}_{(x,Y) \sim \mathcal{D}} \left[ \sum_{i=1}^{T'} \log \sigma \left( \beta \log \frac{\pi_\theta(y_i^w | y_{<i}^w, x)}{\pi_{\text{ref}}(y_i^w | y_{<i}^w, x)} - \beta \log \frac{\pi_\theta(y_i^l | y_{<i}^l, x)}{\pi_{\text{ref}}(y_i^l | y_{<i}^l, x)} \right) \right], \tag{13}$$

*where $\sigma$ is the sigmoid function.*

*Proof.* From the definition of ADPO loss, we have

$$\mathcal{L}_{\text{ADPO}} = -\mathbb{E}_{(x,Y) \sim \mathcal{D}} \left[ \log p_1(y^w \succ y^l | x) \right] \tag{27}$$

$$= -\mathbb{E}_{(x,Y) \sim \mathcal{D}} \left[ \log \prod_{i=1}^{T'} \frac{\exp(-E_1^*(x, y_{\leq i}^w))}{\sum_{y_{\leq i} \in Y_i} \exp(-E_1^*(x, y_{\leq i}))} \right] \tag{28}$$

$$= -\mathbb{E}_{(x,Y) \sim \mathcal{D}} \left[ \sum_{i=1}^{T'} \log \frac{\exp(-E_1^*(x, y_{\leq i}^w))}{\sum_{y_{\leq i} \in Y_i} \exp(-E_1^*(x, y_{\leq i}))} \right] \tag{29}$$

$$= -\mathbb{E}_{(x,Y) \sim \mathcal{D}} \left[ \sum_{i=1}^{T'} \log \frac{1}{1 + \exp(E_1^*(x, y_{\leq i}^w) - E_1^*(x, y_{\leq i}^l))} \right] \tag{30}$$

$$= -\mathbb{E}_{(x,Y) \sim \mathcal{D}} \left[ \sum_{i=1}^{T'} \log \sigma(-(E_1^*(x, y_{\leq i}^w) - E_1^*(x, y_{\leq i}^l))) \right]. \tag{31}$$

where $T' = \max\{\mu'(y^w), \mu'(y^l)\}$. Taking the logarithm of both sides of Eq. (11), we have

$$\log p_2(y_i | y_{<i}, x) = -E_2^*(x, y_{\leq i}) - \log \sum_{y_i \in \mathcal{V}} \exp(-E_2^*(x, y_{\leq i})), \tag{32}$$

and thus we have

$$E_2^*(x, y_{\leq i}) = -\log p_2(y_i | y_{<i}, x) - \log \sum_{y_i \in \mathcal{V}} \exp(-E_2^*(x, y_{\leq i})). \tag{33}$$

From the definition of prefix energies, we obtain

$$E_1^*(x, y_{\leq i}) = \beta \left( E_2^*(x, y_{\leq i}) + \log \pi_{\text{ref}}(y_i | y_{<i}, x) \right) \tag{34}$$

$$= \beta \left( -\log p_2(y_i | y_{<i}, x) - \log \sum_{y_i \in \mathcal{V}} \exp(-E_2^*(x, y_{\leq i})) + \log \pi_{\text{ref}}(y_i | y_{<i}, x) \right) \tag{35}$$

$$= \beta \left( -\log \frac{p_2(y_i | y_{<i}, x)}{\pi_{\text{ref}}(y_i | y_{<i}, x)} \right). \tag{36}$$

Thus, we have

$$E_1^*(x, y_{\leq i}^w) - E_1^*(x, y_{\leq i}^l) = -\beta \log \frac{p_2(y_i^w|y_{<i}^w, x)}{\pi_{\text{ref}}(y_i^w|y_{<i}^w, x)} + \beta \log \frac{p_2(y_i^l|y_{<i}^l, x)}{\pi_{\text{ref}}(y_i^l|y_{<i}^l, x)}. \tag{37}$$

From Eqs. (31) and (37), we obtain

$$\mathcal{L}_{\text{ADPO}} = -\mathbb{E}_{(x,Y)\sim\mathcal{D}} \left[ \sum_{i=1}^{T'} \log \sigma \left( \beta \log \frac{p_2(y_i^w|y_{<i}^w, x)}{\pi_{\text{ref}}(y_i^w|y_{<i}^w, x)} - \beta \log \frac{p_2(y_i^l|y_{<i}^l, x)}{\pi_{\text{ref}}(y_i^l|y_{<i}^l, x)} \right) \right]. \tag{38}$$

When $p_2 = \pi_\theta$, we have

$$\mathcal{L}_{\text{ADPO}} = -\mathbb{E}_{(x,Y)\sim\mathcal{D}} \left[ \sum_{i=1}^{T'} \log \sigma \left( \beta \log \frac{\pi_\theta(y_i^w|y_{<i}^w, x)}{\pi_{\text{ref}}(y_i^w|y_{<i}^w, x)} - \beta \log \frac{\pi_\theta(y_i^l|y_{<i}^l, x)}{\pi_{\text{ref}}(y_i^l|y_{<i}^l, x)} \right) \right]. \tag{39}$$

$\square$

## B  KL-CONSTRAINED REWARD MAXIMIZATION

ADPO leverages the optimal solution of the KL-constrained reward maximization problem. We first review Remark 1 (Peters & Schaal, 2007; Peng et al., 2019; Korbak et al., 2022; Go et al., 2023; Rafailov et al., 2023), and then show Corollary 2.

**Remark 1 (KL-constrained Reward Maximization).** *Given a reward function $r(x, y)$, a reference model $\pi_{\text{ref}}(y|x) > 0$ and a constant $\beta > 0$, we define the objective function for the KL-constrained reward maximization problem as*

$$\mathcal{J}(\pi) = \mathbb{E}_{x\sim\mathcal{D}, y\sim\pi(y|x)} \left[ r(x, y) \right] - \beta D_{KL} \left[ \pi(y|x) || \pi_{\text{ref}}(y|x) \right]. \tag{40}$$

*Then, the optimal solution is given by the Boltzmann distribution induced by the posterior energy:*

$$E_2(x, y) = -\frac{1}{\beta} r(x, y) - \log \pi_{\text{ref}}(y|x). \tag{41}$$

*Proof.* The Boltzmann distribution $p_2$ induced by $E_2$ is given by

$$p_2(y|x) = \frac{\exp(-E_2(x, y))}{Z(x)}, \quad Z(x) = \sum_{y\in\mathcal{Y}} \exp(-E_2(x, y)). \tag{42}$$

Thus, we have

$$\mathcal{J}(\pi) = \mathbb{E}_{x\sim\mathcal{D}, y\sim\pi(y|x)} \left[ r(x, y) \right] - \beta D_{\text{KL}} \left[ \pi(y|x) || \pi_{\text{ref}}(y|x) \right] \tag{43}$$

$$= \mathbb{E}_{x\sim\mathcal{D}, y\sim\pi(y|x)} \left[ r(x, y) - \beta \log \frac{\pi(y|x)}{\pi_{\text{ref}}(y|x)} \right] \tag{44}$$

$$= -\beta \, \mathbb{E}_{x\sim\mathcal{D}, y\sim\pi(y|x)} \left[ \log \exp \left( -\frac{1}{\beta} r(x, y) \right) + \log \frac{\pi(y|x)}{\pi_{\text{ref}}(y|x)} \right] \tag{45}$$

$$= -\beta \, \mathbb{E}_{x\sim\mathcal{D}, y\sim\pi(y|x)} \left[ \log \frac{\pi(y|x)}{\pi_{\text{ref}}(y|x) \exp \left( \frac{1}{\beta} r(x, y) \right)} \right] \tag{46}$$

$$= -\beta \, \mathbb{E}_{x\sim\mathcal{D}, y\sim\pi(y|x)} \left[ \log \frac{\pi(y|x)}{\exp \left( -E_2(x, y) \right)} \right] \tag{47}$$

$$= -\beta \, \mathbb{E}_{x\sim\mathcal{D}, y\sim\pi(y|x)} \left[ \log \frac{\pi(y|x)}{p_2(y|x)} - \log Z(x) \right] \tag{48}$$

$$= -\beta \, \mathbb{E}_{x\sim\mathcal{D}} \left[ D_{\text{KL}} \left( \pi(y|x) || p_2(y|x) \right) - \log Z(x) \right] \tag{49}$$

Then, we obtain

$$\arg\max_\pi \mathcal{J}(\pi) = \arg\min_\pi D_{\text{KL}} \left( \pi(y|x) || p_2(y|x) \right) = p_2 \tag{50}$$

$\square$

**Corollary 2.** *Given a reward function $r(x, y)$ and a reference autoregressive model $\pi_{\mathrm{ref}}(y_i|y_{<i}, x) > 0$ and a constant $\beta > 0$, the following prefix posterior energy induces the optimal solution of the KL-constrained reward maximization problem in Eq. (40):*

$$E_2^*(x, y_{\leq i}) = -\frac{1}{\beta} r^*(x, y_{\leq i}) - \log \pi_{\mathrm{ref}}(y_i|y_{<i}, x), \tag{51}$$

*where $r^*$ is an additive decomposition of $r$.*

*Proof.* Define an energy $\bar{E}_2^*$ as

$$\bar{E}_2^*(x, y) = \sum_{i=1}^{T'} E_2^*(x, y_{\leq i}) \tag{52}$$

Then, we have

$$\bar{E}_2^*(x, y) = -\frac{1}{\beta} \sum_{i=1}^{T'} r^*(x, y_{\leq i}) - \log \prod_{i=1}^{T'} \pi_{\mathrm{ref}}(y_i|y_{<i}, x) \tag{53}$$

$$= -\frac{1}{\beta} r(x, y) - \log \pi_{\mathrm{ref}}(y|x) \tag{54}$$

$$= E_2(x, y). \tag{55}$$

Therefore, we have $\bar{E}_2^* = E_2$. From Remark 1, the Boltzmann distribution induced by $\bar{E}_2^*$ is the optimal solution of the KL-constrained reward maximization problem. $\square$

## C PREFIX-WISE REPARAMETERIZATION COMPLETENESS

**Proposition 1 (Prefix-wise Reparameterization Completeness).** *Let $[r^*]$ denote the reward-shift equivalence class of a prefix-wise reward function $r^* : \mathcal{X} \times \mathcal{Y}^* \to \mathbb{R}$, defined as*

$$[r^*] = \{r' \mid \exists f \ \forall(x, y_{\leq i}) \ r'(x, y_{\leq i}) = r^*(x, y_{\leq i}) + f(x, y_{<i})\}. \tag{56}$$

*Given an autoregressive reference model $\pi_{\mathrm{ref}}(y_i|y_{<i}, x) > 0$ and a hyperparameter $\beta > 0$, for any prefix-wise reward function $r^*$, there exists a unique representative $r_\circ^* \in [r^*]$ such that, for all $x \in \mathcal{X}$ and prefixes $y_{\leq i} \in \mathcal{Y}^*$,*

$$r_\circ^*(x, y_{\leq i}) \equiv \beta \log \frac{\pi(y_i|y_{<i}, x)}{\pi_{\mathrm{ref}}(y_i|y_{<i}, x)}. \tag{57}$$

*for some autoregressive model $\pi$.*

*Proof.* Given a prefix-wise reward function $r^*$, we first show the existence of $\pi$. Define the Boltzmann distribution

$$\pi(y_i|y_{<i}, x) = \frac{\exp(-E_2^*(x, y_{\leq i}))}{Z(x, y_{<i})}, \quad Z(x, y_{<i}) = \sum_{y_i} \exp(-E_2^*(x, y_{\leq i})) \tag{58}$$

Taking logarithms gives

$$\log \pi(y_i|y_{<i}, x) = -E_2^*(x, y_{\leq i}) - \log Z(x, y_{<i}) \tag{59}$$

$$= \frac{1}{\beta} r^*(x, y_{\leq i}) + \log \pi_{\mathrm{ref}}(y_i|y_{<i}, x) - \log Z(x, y_{<i}) \tag{60}$$

Thus, we obtain the required form:

$$r^*(x, y_{\leq i}) = \beta \log \frac{\pi(y_i|y_{<i}, x)}{\pi_{\mathrm{ref}}(y_i|y_{<i}, x)} + \beta \log Z(x, y_{<i}). \tag{61}$$

If another pair $(\tilde{r}, \tilde{\pi})$ enjoys the same property, their log-ratios differ by a function independent of $y_i$, thereby $\tilde{r} - r_\circ^* = f(x, y_{<i})$. Hence both belong to the same shift class, $r_\circ^*$ is unique modulo such shifts. $\square$

# D  EVALUATION METRICS

In this section, we provide detailed descriptions of the evaluation metrics used in our experiments.

## D.1  MATH REASONING TASKS

For GSM8K and MATH500, we report **accuracy**. A prediction is counted as correct if the final numerical answer matches the ground-truth answer after standard normalization (e.g., stripping formatting, ignoring trivial differences).

## D.2  CONVERSATIONAL BENCHMARKS

**AlpacaEval 2** is an automatic preference-evaluation benchmark that uses an LLM-as-a-judge framework to assess instruction-following quality. We report two metrics:

- **Win Rate (WR):** the percentage of pairwise comparisons in which the model's response is preferred over the baseline.
- **Length-Controlled Win Rate (LC):** a debiased variant of WR that adjusts for response-length bias.

**Arena-Hard** is a curated set of difficult prompts from the Chatbot Arena, designed to measure robustness on challenging instruction-following tasks. We report:

- **Win Rate:** the fraction of pairwise comparisons in which the model's response is preferred according to the Arena-Hard judging protocol (LLM-based judges).

**MT-Bench** is a multi-turn dialogue benchmark consisting of 80 conversational questions spanning various categories (e.g., reasoning, coding, safety). Evaluation is conducted with GPT-4-based judges. We report:

- **MT-Bench Score:** the average judge rating on a 1–10 scale across all turns, reflecting helpfulness, coherence, and overall response quality.

# E  ADDITIONAL EXPERIMENTS

In this section, we conduct additional experiments to investigate the impact of several hyperparameters, including the hyperparameter $\beta$ in the objective function, LoRA rank, and model size. We also analyze the learning behavior in the conversational task. The results collectively validate the consistent effectiveness of ADPO across diverse experimental settings and model configurations.

**Hyperparameter $\beta$.** Table 7 provides a hyperparameter analysis of $\beta$ for DPO and ADPO across four LLMs evaluated on GSM8K. ADPO consistently surpasses DPO, demonstrating notable improvements particularly for smaller $\beta$ values (0.5 and 1.0).

**LoRA Rank.** Table 8 compares the effects of different LoRA ranks (4, 16, and 64) on GSM8K for Llama-3-8B. We observe that ADPO outperforms DPO across all ranks and the performance gap widens at higher ranks.

**Model Size.** Table 9 examines the impact of scaling model size from Llama-3-8B to Llama-3-70B on GSM8K. ADPO consistently outperforms DPO at both model sizes, achieving 89.23% with a size of 70B.

Table 7: Hyperparameter study for $\beta$. Results are reported for $\beta = 0.5, 1.0, 1.5$ on GSM8K.

| Method | Llama-3-8B | | | Gemma-3-12B | | | Qwen-3-8B | | | DeepSeek-Math-7B | | |
|--------|------|------|------|------|------|------|------|------|------|------|------|------|
| | 0.5 | 1.0 | 1.5 | 0.5 | 1.0 | 1.5 | 0.5 | 1.0 | 1.5 | 0.5 | 1.0 | 1.5 |
| DPO | 62.47 | 64.37 | 63.91 | 77.71 | 77.03 | 75.82 | 87.87 | 86.96 | 86.73 | 66.94 | 67.78 | 67.63 |
| ADPO | 68.61 | 68.08 | 64.90 | 78.85 | 78.32 | 77.48 | 88.10 | 88.10 | 88.70 | 70.05 | 69.98 | 69.22 |

Table 8: LoRA rank $r$ (Llama-3-8B).

| Method | 4 | 16 | 64 |
|---|---|---|---|
| DPO | 63.53 | 64.37 | 63.84 |
| ADPO | 65.81 | 68.08 | 68.23 |

Table 9: Model Size.

| Method | Llama-3-8B | Llama-3-70B |
|---|---|---|
| DPO | 64.37 | 88.10 |
| ADPO | 68.08 | 89.23 |

Table 10: Full-parameter tuning results (training without LoRA).

| Method | GSM8K | MATH |
|---|---|---|
| DPO | 64.37 | 18.00 |
| ADPO | 67.32 | 20.00 |
| cDPO | 66.19 | 18.00 |
| cADPO | 68.39 | 19.80 |

Table 11: Standard deviations on two mathematical reasoning benchmarks.

| Method | Llama-3-8B | | Gemma-3-12B | | Qwen-3-8B | | DeepSeek-Math-7B | |
|---|---|---|---|---|---|---|---|---|
| | GSM8K | MATH | GSM8K | MATH | GSM8K | MATH | GSM8K | MATH |
| DPO | 0.19 | 0.33 | 0.22 | 0.20 | 0.18 | 0.61 | 0.24 | 0.45 |
| ADPO | 0.19 | 0.33 | 0.22 | 0.32 | 0.06 | 0.73 | 0.29 | 0.41 |
| cDPO | 0.09 | 0.32 | 0.14 | 0.15 | 0.12 | 0.20 | 0.21 | 0.43 |
| cADPO | 0.08 | 0.31 | 0.17 | 0.47 | 0.13 | 0.45 | 0.11 | 0.41 |

Table 12: Average response length.

| Method | Llama-3-8B | | Gemma-3-12B | | Qwen-3-8B | | DeepSeek-Math-7B | |
|---|---|---|---|---|---|---|---|---|
| | GSM8K | MATH | GSM8K | MATH | GSM8K | MATH | GSM8K | MATH |
| DPO | 317.8 | 869.7 | 391.2 | 895.2 | 299.7 | 513.8 | 257.7 | 682.6 |
| ADPO (Ours) | 342.8 | 917.9 | 518.5 | 2192.3 | 294.9 | 525.7 | 280.8 | 662.9 |
| cDPO | 520.4 | 1859.7 | 333.5 | 1174.9 | 299.8 | 526.9 | 261.7 | 696.7 |
| cADPO (Ours) | 308.1 | 847.0 | 287.2 | 841.0 | 313.8 | 510.4 | 250.3 | 630.0 |

**Full parameter tuning.** Table 10 provides the results of full-parameter tuning (*i.e.* without LoRA) for Llama-3-8B. ADPO and cADPO still outperform DPO and cDPO even under full-parameter training.

**Standard Deviation.** Table 11 reports the standard deviations corresponding to the results in Table 3 of the main paper. The standard deviations are consistently small (typically below 1%), and the relative ranking of methods remains unchanged, confirming that our conclusions are robust to evaluation noise.

**Generation Length.** Table 12 reports the average output length across all models and datasets. We observed two trends: (a) ADPO produces slightly longer responses than DPO, while (b) cADPO produces shorter responses than cDPO.

ADPO performs preference comparison at each prefix, encouraging the model to accumulate positive evidence step-by-step. This finer-grained signal encourages the model to articulate intermediate reasoning more explicitly, which naturally results in slightly longer outputs.

cDPO amplifies penalties for "critical tokens" at the sequence level. This can lead to longer outputs, because producing additional reasoning steps gives the model more chances to place tokens that are not penalized. In contrast, cADPO applies the same penalties locally at each prefix, preventing such length-driven attenuation. As a result, cADPO discourages unnecessary elaboration and yields more concise outputs.

**Learning Behavior.** Figure 3 shows training dynamics for the conversation task by plotting the evolution of sequence log probabilities over training steps for DPO and ADPO. In the static family

(left), reducing the window size increases the margin between preferred (solid lines) and dispreferred (dashed lines) trajectories, showing greater separation compared to DPO. In the adaptive family (right), increasing the number of segments similarly amplifies this margin. Across both settings, ADPO consistently achieves a clearer divergence between preferred and dispreferred responses throughout training.

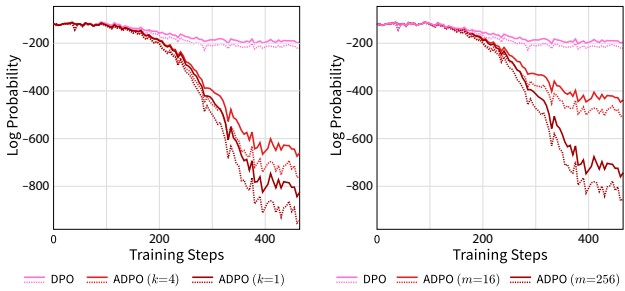

Figure 3: Training dynamics (conversation task).

**Implementation Details.** We implemented our methods using the Transformer Reinforcement Learning (TRL) library. For training on GSM8K, we followed the experimental setup of Lin et al. (2025). Specifically, positive and negative models are first trained for 1 epoch with a learning rate of $3 \times 10^{-4}$. Subsequently, preference optimization models are trained with $\beta = 1.0$ for 3 epochs using AdamW with LoRA ($r = 16$), at a learning rate of $2 \times 10^{-5}$ for DPO and $4 \times 10^{-5}$ for cDPO. Each problem was sampled 64 times with a top-p probability of 50% during contrastive estimation. For the conversation task, all models are trained for 1 epoch using preference data scored by PairRM. Experiments are conducted using four NVIDIA H100 GPUs. We will release our code and models.

# F ADDITIONAL ANALYSIS

## F.1 PREFIX-WISE REWARD DYNAMICS

As formalized in Eq. 15, ADPO assigns an implicit reward at each step $i$. To understand how these implicit rewards are distributed across prefixes and how they evolve during training, we analyze two summary statistics of the prefix-wise reward signal: (A) the reward variance, which reflects how stable or uncertain the reward is at each prefix, and (B) the reward margin (chosen–rejected difference), which captures where preference differences are realized. We normalize prefix positions to $[0, 1]$ and evaluate these quantities at early, middle, and late checkpoints from ADPO training on the Llama-3-8B model for the math reasoning task. Figures 4 (A) and (B) illustrate how the variance and margin profiles evolve across training.

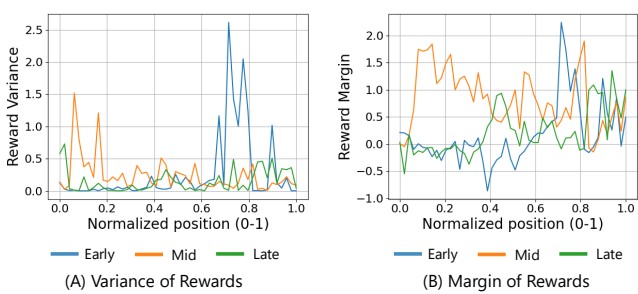

Figure 4: Prefix-wise reward variance and margin.

**Early Training.** As shown in the blue curves in Figures 4 (A) and (B), both reward variance and margin are concentrated in the middle-to-late prefix region. This indicates that ADPO first focuses on correcting the unstable reasoning steps that occur near the end of the response, where chosen and rejected outputs differ most and the policy is initially most uncertain.

**Middle Training.** In the mid-stage checkpoints (orange curves), the mass of both variance and margin shifts toward earlier prefixes. Once the later reasoning stabilizes, ADPO begins refining the early parts of the reasoning chain: these early steps determine how the model sets up the solution and whether the subsequent reasoning flows toward a correct final answer.

**Late Training.** In the late-stage checkpoints (green curves), reward variance becomes small across all prefixes, while the margin becomes sharply concentrated in the final portion of the sequence. This suggests that the global reasoning structure has largely converged, and ADPO focuses on fine-tuning the last few steps that directly determine correctness.

**Summary.** Overall, these results reveal a consistent three-phase progression: ADPO first improves the end of the reasoning path, then the beginning, and finally sharpens the answer-producing tokens

themselves. This demonstrates that ADPO performs meaningful prefix-wise credit assignment, progressively localizing the implicit reward to the prefixes that matter most for preference.

