# OpenReview forum: "Autoregressive Direct Preference Optimization"
_ICLR.cc/2026/Conference — Submitted to ICLR 2026_

### Official Review · Reviewer_cRUn · 2025-10-29

**Soundness:** 3
**Presentation:** 2
**Contribution:** 3
**Rating:** 2
**Confidence:** 4

**Summary:**

Although DPO have achieved an effective way of training LLMs to align with human preferences, the preference signal of DPO is compared at the sentence level. Therefore, the autoregressive property is represented after fully generating the reponses. This paper suggests a new method to explicitly represent the autoregressive propoerty to the bradley yerry model. This paper theoretically show every reward model can be reformulated as the autoregressive version. and experimentally show improved performances over the baselines.

**Strengths:**

- Another trial to decompose the reward calculation from sentence to token.
- Theoretical consistency between reward modeling and autoregressive decoding is well-motivated.

**Weaknesses:**

- Additive decomposition may not be true because it is a sentence. There may be some correlation between words.
- Experiment is not enough. Both GSM8K and MATH500 are limited to math reasoning tasks. Why not more general human preference dataset, e.g. UltraFeedback? I suppose math task is not adequate to show the granularity consideration is effective, because after all the ultimate objective of math task is to solve the questions with explicit answer. I also saw there is "conversation tasks" section. What did the authors do in this section? Training on mah data and test on conversation benchmark?
- Only the naive DPO, cDPO and SimPO are compard as baselines if I count for the whole experiments; yet some were only partially experimented. Why no results of cDPO for Table5?, Why no results of SimPO for table3?
- No explanation on evaluation metric. I know what those are, but it severely compromises the quality of the paper.

**Questions:**

- What can be an intuitive explanation of ADPO to maximize the log likelihood ratio of winning response's token and losign response;s token at a specific time t? I understood this means finer granularity whether than taking the whole responses just as 1. However, the reason why previous researches took the total chunk of responses as 1 was because a response will finally have a complete meaning only a reponse has ended. Therefore, taking just a word difference of same time between a positive response and a negative response may not have a specific meaning. This signal becomes more harder to understand when the format of winning response and losing response becomes more differernt.
- What do the authors mean by "In Table 6, we also find that granularity exhibits a similar trend as in mathematical reasoning tasks:" ? Table 6 is the result of AlpacaEval2?
- With so unaligned experimental results section as above, how can I trust the reproducibility of the experimental results at all?
- Model performance of training without LoRA?
- I am willing to change my opinion to accept if my concerns with regard to the experiment parts are well explained.

---

> ### Author Response · Authors · 2025-11-25
> **Rebuttal by Authors (1)**
>
> Thank you very much for your insightful review and comments. Below, we respond to each point in detail.
>
> > Weakness1: Additive decomposition may not be true because it is a sentence. There may be some correlation between words.
>
> Importantly, our formulation does not assume that natural-language sentences are additive or that tokens are independent. The additive decomposition in Definition 3 is not an assumption about linguistic structure, but the general form that any reward function can always be equivalently rewritten into, as guaranteed by Lemma 1 and Theorem 1 in our paper.
>
> Therefore, this decomposition does not restrict modeling capacity nor imply token-wise independence; it simply expresses an arbitrary reward in an equivalent prefix-wise form without altering any correlations among words.
>
> > Weakness2: Both GSM8K and MATH500 are limited to math reasoning tasks. Why not more general human preference dataset, e.g. UltraFeedback?
>
> In the “Conversation Tasks” section, we train models on UltraFeedback and evaluate them on AlpacaEval 2, Arena-Hard, and MT-Bench.
> This training data description was missing from the original submission, and we have now added it explicitly.
>
> These results demonstrate that the benefits of ADPO’s finer granularity extend beyond mathematical reasoning to general human-preference tasks as well.
>
> > Weakness3: Why no results of cDPO for Table5?, Why no results of SimPO for table3?
>
> The table below reports SimPO results in Table 3. As shown, ADPO and cADPO outperform SimPO across all settings:
>
> |  | Llama-3-8B, GSM8K | Llama-3-8B, MATH | Gemma-3-12B, GSM8K | Gemma-3-12B, MATH | Qwen-3-8B, GSM8K | Qwen-3-8B, MATH | DS-Math-7B, GSM8K | DS-Math-7B, MATH |
> | --- | --- | --- | --- | --- | --- | --- | --- | --- |
> | DPO | 64.37 | 18.00 | 77.03 | 39.80 | 86.96 | 53.80 | 67.78 | 32.00 |
> | ADPO | 68.08 | 21.00 | 78.32 | 41.20 | 88.10 | 55.40 | 69.98 | 33.40 |
> | cDPO | 67.90 | 16.80 | 77.18 | 38.60 | 90.98 | 56.80 | 72.90 | 33.40 |
> | cADPO | 68.76 | 20.20 | 78.85 | 40.40 | 91.74 | 57.20 | 73.54 | 35.40 |
> | SimPO  | 62.32  | 19.20  | 77.26  | 39.80  | 87.19  | 55.20  | 68.84  | 32.80 |
>
> cDPO requires explicit True/False labels for each response, which are available for math datasets but not for UltraFeedback.
> Therefore, cDPO cannot be applied to conversational tasks and cannot be included in Table 5.
>
> > Weakness4: No explanation on evaluation metric.
>
> We have added clear definitions of all evaluation metrics in Appendix D.
> A concise summary is provided below.
>
> - Math reasoning tasks:
>     - Accuracy:  A prediction is considered correct if the final normalized numerical answer matches the ground truth. For MATH500, we use sympy to parse and evaluate LaTeX-formatted answers.
> - Conversational tasks:
>     - AlpacaEval 2: A LLM-as-a-Judge–based preference benchmark for instruction following.
>       - Win Rate (WR): Percentage of pairwise comparisons where the model’s answer is preferred.
>       - Length-Controlled Win Rate (LC): A debiased version of WR that accounts for response-length bias.
>     - Arena-Hard: A curated subset of difficult prompts from Chatbot Arena.
>       - Win Rate: Fraction of pairwise comparisons where the model’s response is preferred under the Arena-Hard judging protocol.
>     - MT-Bench: A multi-turn dialogue benchmark scored by GPT-4–based judges over 80 questions.
>       - The reported metric is the average judge rating (1–10).
>
> > Question1: What can be an intuitive explanation of ADPO to maximize the log likelihood ratio of winning response's token and losing response’s token at a specific time t? Taking just a word difference of same time between a positive response and a negative response may not have a specific meaning.
>
> First, ADPO is not a heuristic that tries to maximize step-level likelihood differences. The per-prefix form of the loss arises automatically when we fix the theoretical inconsistency in DPO, specifically, making the Boltzmann posterior consistent with an autoregressive model. Once this mismatch is resolved, the prefix-wise BT model and the ADPO objective follow directly from the derivation, not from any token-level assumption.
>
> If we seek intuition for such step-level maximization, it reflects how the model shifts its probability mass at that prefix relative to the reference model, indicating whether the local decision moves the trajectory toward or away from the preferred response.
>
> > Question2: What do the authors mean by "In Table 6, we also find that granularity exhibits a similar trend as in mathematical reasoning tasks:" ? Table 6 is the result of AlpacaEval2?
>
> Yes. Table 6 reports AlpacaEval 2 results for different granularity settings of ADPO (varying k and m).
>
> As described in Section 5.3 and shown in Figure 1:
> - smaller k → finer granularity (static family)
> - larger m → finer granularity (adaptive family)
>
> Table 6 shows that finer granularity again improves performance, mirroring the same trend observed for math reasoning in Table 4.

---

> ### Author Response · Authors · 2025-11-25
> **Rebuttal by Authors (2)**
>
> > Question4: Model performance of training without LoRA?
>
> The table below compares DPO/ADPO and cDPO/cADPO under full-parameter (non-LoRA) training on Llama-3-8B for math reasoning tasks.
> ADPO and cADPO still outperform their DPO counterparts, confirming that our findings are not specific to LoRA.
>
> |  | Llama-3-8B, GSM8K | Llama-3-8B, MATH |
> | --- | --- | --- |
> | DPO | 64.37 | 18.00 |
> | ADPO | 67.32 | 20.00 |
> | cDPO | 66.19 | 18.00 |
> | cADPO | 68.39 | 19.80 |
>
> Note that all conversational-task experiments were already conducted without LoRA.

---

### Official Review · Reviewer_waf7 · 2025-10-31

**Soundness:** 2
**Presentation:** 3
**Contribution:** 3
**Rating:** 4
**Confidence:** 3

**Summary:**

In this paper, a reformulation of the standard DPO framework, called Autoregressive Direct Preference Optimization (ADPO) is introduced. The core problem of DPO is the application of the Bradley-Terry model at the level of complete responses, and when the autoregressive property of the language model is considered only after the target response is defined. ADPO resolves this issue by explicitly incorporating the autoregressive assumption before the application of the BT model.

The authors achieve this by applying the domain of energy functions based on the reward signal from the output space $\mathcal{Y}$ to the prefix space $\mathcal{Y}^{*}$. Specifically, the prefix-wise BT (Equation 10) model is developed, which determines the probability of the preference as the product of preferences over the prefixes.Theoretically, the authors prove the reparameterization completeness and distinguish the difference between two length metrics: token length $\mu$ and feedback length $\mu’$. The case of $\mu’=1$ corresponds to DPO. Additionally, two families for implementation are proposed – Static and Adaptive. Empirically, the authors demonstrate improvements on mathematical and conversational datasets, simultaneously, for the DPO and other competitive methods (cDPO, SimPO).

**Strengths:**

1. The observation that standard DPO amounts to applying the BT model at the response level, singling out the autoregressive structure after deriving the objective, is a significant theoretical mismatch in the existing formalization. ADPO rectifies this issue by introducing an energy definition over the prefix closure and explicitly assuming an autoregressive reference model at that point. This theory is much closer to the reality of how an LLM actually generates text.
2. It is clearly shown that how they used a Boltzmann distribution to re-formulate the foundations of DPO in Section 3 while providing the derivation of the ADPO loss in Appendix A makes for a robust argument. Another theoretical contribution is the introduction of the two length measures in Section 5.3.
3. In Section 5.4, the introduced "Granularity Families" offers guidelines for exploring the continuous optimization space from token-level to response-level. Further on, using "strong composition" for the definition of the mapping from tokens to concepts to subsequences is a reasonable decision.
4. ADPO, especially at finer granularities, achieves a more pronounced separation between the log probabilities of preferred and dispreferred sequences during training than DPO. This suggests that improved "information per step" is a correct intuition for ADPO, and its inspiration from the early-dominant-prefix guidance in prefix-wise preference training is accurate.

**Weaknesses:**

1. About the implicit reward function $r^*(x, y_{\le i})$: In the standard DPO, the implicit reward has an explanation related to the complete response. The optimization in ADPO is tied to the delivered localized rewards. What these rewards could be? Have they a variance? How do they ‘credit’ prefixes for being ‘good’ or ‘bad’ in the response?
2. About effect of moving the summation outside the log-sigmoid function that changes the non-linearity: we know that DPO computes the total advantage, the sum of log-ratios, and then applies the sigmoid function once. Thus, DPO is sensitive to the total margin, but the sigmoid collapses all but very large margins. In contrast, ADPO fully applies the sigmoid to the advantage of each token or subsequence and only then sums the results. But such a change in the non-linearity will change the learning dynamics. For example, it becomes much more challenging to tell a sequence if a cut cost is very high but concentrated on one token is worse than a sequence where the same total costs are spread across many tokens after accumulation because the local advantages are now squashed before accumulating. This alteration affects optimization and regularization, which need further discussion.
3. For experiment: The motivation for ADPO is a drive for finer-grained optimization, this is not unique: it is the same as the motivation for token-level Reinforcement Learning from Human Feedback. Therefore, the paper should contain comparison with an optimized implementation of token-level PPO. The claim is that DPO is generally more efficient than PPO. While this may be the case for ADPO, you can further establish the relationship between ADPO and token-level RL optimization.
4. About Static Family: Section 5.4 introduces the necessity for EOS padding to make the lengths of the preferred and dispreferred responses equal before the decomposition into subsequences. The authors themselves acknowledge that this is a critical restriction, writing that EOS padding “potentially constrains its performance.” Furthermore, deriving sub-transformers from substrings that have been distorted by optimizing over padded tokens introduces noise and inefficiencies in the form of an indistinguishable prefix problem. It seems like the Adaptive Family is a more viable approach?

**Questions:**

1. About prefix-wise BT model (Eq. 10): independence assumption: Human preference can often not be expressed by decomposable into independent prefix preference. Did you experiment for cases where this assumption is skeptic, such as the task where correctness/preference is only controlled by the target of the prefixes, i.e., the final answer in a math problem, and how does ADPO perform in such cases compared to DPO?
2. Also refer to [weakness 2]. And how does this influence the weights of training examples, particularly comparing to short and long sequences? Does this implicitly add the length normalization or regularization different than DPO or SimPO?  it is also fine to explain something about weakness 2.
3. Any analysis on the implicit prefix reward learned by ADPO? I am wondering about the behavior of these reward values that are localized. How do the variance and magnitude of those prefix rewards change during training, and how is the credit distribution learned to the sequence compared to the DPO’s implicit reward?
4. Also, refer to [weakness 4]. For that, it is okay to provide some explanation as well. Or did you come up with an alternative that might combine the benefit of both approaches or other ways to handle the variable lengths or defying the subsequence, perhaps by the user’s natural semantic instead of length?
5. Does  ADPO (at the finest granularity) introduce any computational overloads compared to DPO?

---

> ### Author Response · Authors · 2025-11-25
> **Rebuttal by Authors (1)**
>
> Thank you very much for your constructive review and feedback. Below, we respond to each question and comment.
>
> # Response to Weakness1 and Question 3: Implicit reward in ADPO
>
> > Weakness1: What the implicit rewards in ADPO could be?
>
> As described in Eq. (15), the implicit reward in ADPO is the prefix-wise log-likelihood ratio between the learnable model and the reference model:
>
> $r^*(x, y_{\le i})
> \equiv
> \beta \log \frac{\pi_\theta(y_i | y_{<i}, x)}
> {\pi_{\mathrm{ref}}(y_i | y_{<i}, x)}.$
>
> This quantity has a clear probabilistic interpretation:
>
> - it measures how much the model improves or degrades its likelihood at the specific prefix relative to the reference policy, and
> - it corresponds exactly to the local per-token advantage under KL-regularized RLHF.
>
> Crucially, this reward is not arbitrarily assumed. It is the unique prefix-wise reward implied by the autoregressive factorization of the Boltzmann distribution and by Theorem 1, which establishes the reparameterization completeness of ADPO.
>
> > How do they ‘credit’ prefixes for being ‘good’ or ‘bad’ in the response?
>
> Because ADPO applies the Bradley–Terry model at every prefix, the preference signal is allocated to the exact locations where the preferred response has a higher likelihood ratio than the rejected one. This yields:
>
> - token-level credit assignment without training a separate reward model, and
> - consistency with RLHF, where log π − log π_ref is the canonical per-token advantage signal.
>
> Importantly, ADPO does not impose an artificial assumption that humans evaluate prefixes independently. Rather, the prefix-wise BT structure derives the per-prefix comparisons as the only decomposition compatible with KL-constrained reward maximization (Appendix B).
>
> > Question3: Any analysis on the implicit prefix reward learned by ADPO? How do the variance and magnitude of those prefix rewards change during training, and how is the credit distribution learned to the sequence compared to the DPO’s implicit reward?
>
> To better understand how ADPO’s implicit prefix reward evolves, we analyzed the variance and margin (chosen–rejected difference) across prefix positions during training (See Figure 4 in Appendix F in the revised manuscript.)
> The results reveal a clear three-phase pattern:
>
> 1. Early phase.
>
> Reward variance and margin are concentrated in the mid-to-late prefixes, indicating that ADPO first corrects the later reasoning steps where chosen and rejected responses differ most.
>
> 2. Mid phase.
>
> As training progresses, these signals shift toward earlier prefixes, showing that ADPO begins refining the initial setup and intermediate reasoning steps once the later parts stabilize.
>
> 3. Late phase.
>
> Variance becomes small across all positions, while the margin is focused on the final answer tokens, suggesting convergence of global reasoning structure with fine-grained adjustments near the output.
>
> Overall, ADPO allocates credit in a structured manner, progressing from late reasoning to early reasoning and ultimately to answer tokens, demonstrating meaningful prefix-wise credit assignment.

---

> ### Author Response · Authors · 2025-11-25
> **Rebuttal by Authors (2)**
>
> # Response to Weakness2 and Question2: Characteristic of ADPO
>
> > Weakness2: it becomes much more challenging to tell a sequence if a cut cost is very high but concentrated on one token is worse than a sequence where the same total costs are spread across many tokens
>
> We interpret and formalize this comment as follows.
> Consider two cases:
> - Case A: a single extremely large margin (an, 0, 0, …, 0)
> - Case B: the same total margin spread evenly (a, a, …, a)
>
> The reviewer suggests that applying the sigmoid at each prefix may make ADPO less sensitive to the difference between these two scenarios.
> However, our analysis indicates that ADPO in fact distinguishes these scenarios more effectively.
>
> DPO aggregates the margins before the sigmoid:
>
> $L_{\mathrm{DPO}} = -\log\sigma\left(\sum_i\Delta_i\right).$
>
> Because both A and B have the same total margin an, this yields:
>
> $L_{\mathrm{DPO}}(A)=L_{\mathrm{DPO}}(B).$
>
> Thus, DPO cannot distinguish the two cases at all.
>
> However, ADPO applies the sigmoid to each prefix before summation:
>
> $L_{\mathrm{ADPO}}(\Delta)
> =\sum_i \ell(\Delta_i),
> \text{where }\ell(x)=-\log\sigma(x)=\log(1+e^{-x}).$
>
> The function $\ell(x)$ is convex, so by Jensen’s inequality:
>
> $\frac{1}{n}\big[\ell(an)+(n-1)\ell(0)\big]
> \\ge\
> \ell(a).$
>
> Multiplying by n:
>
> $\ell(an)+(n-1)\ell(0)
> \>\
> n\ell(a),$
>
> Thus,
>
> $L_{\mathrm{ADPO}}(A) > L_{\mathrm{ADPO}}(B).$
>
> Thus, **ADPO penalizes a catastrophic local error more strongly, whereas DPO cannot distinguish the two cases.**
>
> > Question2: (Related to Weakness2) length-normalization and example weighting
>
> The reviewer also asks how ADPO affects the effective weighting of long and short sequences.
>
> We measured the average output length across all models and datasets and consistently observed two trends:
>
> (a) ADPO produces slightly longer responses than DPO, while (b) cADPO produces shorter responses than cDPO.
>
> These trends can be interpreted as follows.
>
> (a) Why ADPO > DPO in length?
>
> ADPO performs preference comparison at each prefix, encouraging the model to accumulate positive evidence step-by-step.
> This finer-grained signal promotes clearer intermediate reasoning, naturally resulting in slightly longer outputs.
>
> (b) Why cADPO < cDPO in length?
>
> cDPO amplifies penalties for “critical tokens” at the sequence level.
> This can encourage longer outputs, as additional tokens offer opportunities to dilute these penalties.
> In contrast, cADPO applies the same penalties locally at each prefix, preventing this dilution.
> As a result, cADPO discourages unnecessary elaboration and yields more concise outputs.
>
> |  | Llama-3-8B, GSM8K | Llama-3-8B, MATH | Gemma-3-12B, GSM8K | Gemma-3-12B, MATH | Qwen-3-8B, GSM8K | Qwen-3-8B, MATH | DS-Math-7B, GSM8K | DS-Math-7B, MATH |
> | --- | --- | --- | --- | --- | --- | --- | --- | --- |
> | DPO | 317.8 | 869.7 | 391.2 | 895.2 | 299.7 | 513.8 | 257.7 | 682.6 |
> | ADPO | 342.8 | 917.9 | 518.5 | 2192.3 | 294.9 | 525.7 | 280.8 | 662.9 |
> | cDPO | 520.4 | 1859.7 | 333.5 | 1174.9 | 299.8 | 526.9 | 261.7 | 696.7 |
> | cADPO | 308.1 | 847.0 | 287.2 | 841.0 | 313.8 | 510.4 | 250.3 | 630.0 |
>
> # Response to Weakness3: Comparison with token-level PPO
>
> > Weakness3: Therefore, the paper should contain comparison with an optimized implementation of token-level PPO. The claim is that DPO is generally more efficient than PPO. While this may be the case for ADPO, you can further establish the relationship between ADPO and token-level RL optimization.
>
> As this paper aims to extend the theoretical foundations of DPO, we believe that verifying the efficiency of DPO against PPO is out of scope.
> Our research question is how the Bradley–Terry model should be applied when the underlying policy is autoregressive, which is fully addressed by the comparisons among DPO, ADPO, and cDPO, a state-of-the-art token-level DPO variant.
>
> Furthermore, there are technical and conceptual difficulties in comparing ADPO with token-level PPO. The only existing work explicitly proposing token-level PPO is TPPO[1]. However, TPPO was introduced specifically for query generation, not preference optimization, and its official implementation repository is currently empty (https://anonymous.4open.science/r/TPPO-D6C6). Thus, a direct empirical comparison would neither be feasible nor methodologically fair.
>
> [1] Token-level Proximal Policy Optimization for Query Generation, Ouyang et al. EMNLP2025.

---

> ### Author Response · Authors · 2025-11-25
> **Rebuttal by Authors (3)**
>
> # Response to Weakness4 and Question4: Segmentation and Adaptive Family
>
> > Weakness4: EOS padding is needed to make the lengths of the preferred and dispreferred responses equal before the decomposition into subsequences.  … It seems like the Adaptive Family is a more viable approach?
>
> We agree with the reviewer’s observation:
> the Static Family requires EOS padding, and this padding can introduce distortions that may limit performance.
>
> Indeed, our experiments on Table 4 clearly show that **Adaptive ADPO > Static ADPO**, confirming that the adaptive decomposition avoids these padding artifacts.
>
> However, we emphasize that this is not a weakness of ADPO itself.
> Despite the padding issue, **every Static ADPO variant still outperforms vanilla DPO.**
> Thus, both Static and Adaptive Families are effective, while the Adaptive Family achieves the best results simply because it eliminates the EOS-padding constraint.
>
> > Question4: did you come up with an alternative that might combine the benefit of both approaches or other ways to handle the variable lengths or defying the subsequence, perhaps by the user’s natural semantic instead of length?
>
> We acknowledge that other definitions of the feedback-length measure $\mu’$ are possible and potentially interesting.
> For example, mathematical reasoning could use step-level decomposition, and conversational tasks could use semantic units, as the reviewer suggests. However, since the Adaptive approach already provides robust performance without additional heuristics, exploring alternative segmentations is beyond the scope of this paper and left for future work.
>
> # Response to Question1: Math problem
>
> > Question1: Did you experiment for cases where this assumption is skeptic, such as the task where correctness/preference is only controlled by the target of the prefixes, i.e., the final answer in a math problem, and how does ADPO perform in such cases compared to DPO?
>
> Although math tasks ultimately depend on the final answer, real model outputs include full reasoning chains.
> Thus preferences are not determined solely by the final answer.
> Since ADPO can exploit prefix-level distinctions in reasoning quality, it performs well even in such settings. Indeed, our main experiments on GSM8K and MATH500 (Table 3) show that ADPO and cADPO consistently outperform DPO and cDPO.
>
> # Response to Question5: Computational cost
>
> > Question5: Does ADPO (at the finest granularity) introduce any computational overloads compared to DPO?
>
> We provide wall-clock training times (seconds) for all granularities.
>
> The differences between DPO and ADPO are within 100 seconds, showing that ADPO introduces no meaningful computational overhead, even at the finest granularity.
>
> |  | Llama-3-8B |
> | --- | --- |
> | ADPO (static, k=4) | 2314 |
> | ADPO (static, k=2) | 2228 |
> | ADPO (static, k=1) | 2301 |
> | ADPO (adaptive, m=1) (equivalent to DPO) | 2234 |
> | ADPO (adaptive, m=2) | 2287 |
> | ADPO (adaptive, m=16) | 2321 |
> | ADPO (adaptive, m=128) | 2318 |
> | ADPO (adaptive, m=256) | 2226 |
> | ADPO (adaptive, m=512) | 2327 |

---

### Official Review · Reviewer_mCzu · 2025-10-31

**Soundness:** 3
**Presentation:** 3
**Contribution:** 3
**Rating:** 6
**Confidence:** 4

**Summary:**

The paper derives an autoregressive variant of DPO where rewards are assigned not only to complete responses but also prefixes which results in ADPO where the summation and log-sigmoid are swapped. Variants based on different way to dividing the responses are also tested. They demonstrate improved performance over DPO and other baselines on both math reasoning datasets as well as conversational datasets.

**Strengths:**

Novel objective - The paper considers deriving the objective building in an assumption of having autoregressive models. They demonstrate that the resulting objective still achieves the optimal policy and also demonstrates that DPO is a special case of the generalized ADPO objectives. They also provide a thorough analysis of the resulting objective and present it in a clear way.

Experimental results - Experimental results demonstrate consistent improvement across multiple tasks and benchmarks and also study the effect of different factors. Details on datasets, models, baselines, and implementation as well as additional exploration of hyperparameters are provided and clear, further supporting the benefits of ADPO.

**Weaknesses:**

Impact - While the benefit of ADPO is clear, one thing that is unclear from the paper is why the model being autoregressive and the output distribution not being autoregressive is considered to be mismatched and why it is expected to be an issue. There are different ways of chunking the outputs as seen with the variants of ADPO, so it seems unclear why treating the output as a single object should lead to issues. I think further reasoning here being provided would strengthen the paper and make the impact of the paper more clear beyond exploring a new dimension that the DPO objective can be changed along.

**Questions:**

- Why is the model being autoregressive and the output distribution not being autoregressive considered to be mismatched, and why is it expected to be an issue?

Addressing this question and clarifying this in the paper would significantly help.

---

> ### Author Response · Authors · 2025-11-25
> **Rebuttal by Authors**
>
> Thank you very much for your constructive review and positive feedback.
>
> > Why is the model being autoregressive and the output distribution not being autoregressive considered to be mismatched?
>
> DPO defines the target distribution $p_2(y|x)$ as a sequence-level Boltzmann distribution (Eq. 4), which does not factorize autoregressively, whereas LLMs necessarily represent distributions of the form
> $\pi_\theta(y|x)=\prod_i \pi_\theta(y_i|y_{<i},x).$
>
> Thus, the optimal distribution induced by DPO generally lies outside the autoregressive family that the model can express. In practice, this means that DPO implicitly specifies a target with a factorization structure different from that of the model, requiring the optimizer to project a non-autoregressive distribution onto an autoregressive parameterization.
>
> > Why is it expected to be an issue?
>
> This mismatch leads to concrete optimization challenges. Because DPO aggregates all token-level advantages into a single margin before applying the sigmoid, prefix-level distinctions become indistinguishable, making the objective insensitive to where errors occur within the sequence. In addition, changes to a single token’s likelihood can influence the global Boltzmann probability in non-local ways, producing gradients that are harder to interpret and less aligned with the autoregressive decomposition of the model.
>
> Since the true optimum of the DPO objective typically does not lie within the autoregressive family, the model is effectively optimizing toward a distribution it cannot represent exactly, which can reduce stability and learning efficiency. ADPO addresses this issue by inducing an autoregressive target distribution that preserves prefix-level preference information and matches the model’s factorization structure. This improves both alignment between the target and model class and the locality of gradient signals.

---

### Official Review · Reviewer_PGZo · 2025-11-03

**Soundness:** 3
**Presentation:** 2
**Contribution:** 2
**Rating:** 4
**Confidence:** 4

**Summary:**

The paper proposes a variant of Direct Preference Optimization (DPO), called Autoregressive DPO (ADPO). The origin of DPO is reformulated using two Boltzmann distributions with reward-based energies defined over the output space $\mathcal{Y}$. By extending the energy domain from $\mathcal{Y}$ to $\mathcal{Y}^*$, the energy definition with the prefix-wise BT model is achieved. One of the biggest differences of ADPO from the original DPO is that the summation across the tokens is moved outside the log-sigmoid function, allowing training with higher granularities. The paper presents several experiment results showing how the performance of ADPO changes along the change of granularity applied to the training process, while ADPO shows significant improvement from the original DPO at its optimal ranges of granularity.

**Strengths:**

1. Theoretical soundness: The paper effectively explains how a different perspective in the DPO formulation can lead to token-wise interpretation, which also allows having a new loss with controllable granularity.
2. The paper is well-organized in terms of logical flow and visual presentation.

**Weaknesses:**

1. I think the biggest issue of this paper is that it is not citing and comparing with a very similar analysis from the authors of the original DPO paper [1]. While there are a few differences in terms of formulation or notations, [1] also present a token-level perspective of the DPO formulation and that token-level DPO can parameterize any dense reward function (section 4.2 of [1]).
2. While the paper is presenting several results where ADPO outperforms DPO, it seems it is still lacking a reasonable explanation why ADPO is supposed to perform better than DPO.
3. The experiment results are not showing standard deviation of the test performances, which are crucial for reinforcing the credibility of the results.

[1] "From $r$ to $Q^*$: Your Language Model is Secretly a Q-Function", Rafailov et al., 2024.

**Questions:**

## Questions
1. What could be the reason ADPO is performing better than the original DPO? Is ADPO basically generating more training signals than DPO, given the same amount of prompts?
2. Do ADPO and DPO exhibit similar wall clock time for training? Does the training time of ADPO change along the change of granularity level?
3. How does ADPO affect the overall response length, compared to other algorithms?

## Suggestions
1. Like I mentioned in the Weakness section, I think it is crucial to cite (Rafailov et al., 2024) and compare the contribution of ADPO's paper with it.
2. Figure 2: I think it is better to use the same color for the same algorithm across the plots.
3. I think presenting the performances of the base model, without any fine-tuning, in the experiment results will benefit the paper.

---

> ### Author Response · Authors · 2025-11-25
> **Rebuttal by Authors (1)**
>
> Thank you very much for your insightful review and feedback. Below, we respond to each question and comment.
>
> > Weakness1, Suggestion1: Cite and show the relation with [1]
>
> We fully agree that this work provides an important token-level analysis of DPO. We have now incorporated [1] into the Related Work section in the revised manuscript.
>
> Importantly, while [1] and our work both involve token-level perspectives of DPO, their goals and contributions differ fundamentally:
>
> - Different objectives:
>     - [1] analyzes DPO through the lens of the token-level MDP and shows that DPO implicitly learns an optimal soft Q-function for some reward. The focus of [1] is to characterize the implicit reward, credit assignment structure, and the connection to soft Q-learning.
>     - In contrast, ADPO does not reinterpret DPO, but **modifies the Bradley–Terry model itself by extending the energy domain from the response space $Y$ to its prefix closure $Y^{*}$**. This leads to a prefix-wise BT model and a new DPO objective, where the summation naturally moves outside the log-sigmoid.
> - Different theoretical outcomes:
>     - While [1] proves that DPO can represent any dense reward via Q-functions, our work proves prefix-wise reparameterization completeness, introduces two distinct length measures ($\mu$, $\mu’$), and shows that DPO is a special case of ADPO when $\mu’$=1. These results are orthogonal to those in [1].
>
> > Weakness2, Question1: What could be the reason ADPO is performing better than the original DPO? Is ADPO basically generating more training signals than DPO?
>
> ADPO performs better than DPO because it aligns the preference model with the autoregressive structure of LLMs, preserving prefix-level preference information that DPO collapses at the sequence level, as described in Section 4.
>
> More practically, in DPO, all token-level advantages are summed before a single sigmoid is applied (see Eq. 6), which compresses local preference differences into a single scalar margin. This makes DPO insensitive to where the model performs well or poorly within a sequence.
>
> In contrast, ADPO extends the Bradley–Terry model to the prefix closure, producing an autoregressive target distribution that matches the factorization of LLMs. As a result, the objective applies the sigmoid at each prefix, preserving meaningful local preference information.
>
> Thus, the answer to “Is ADPO basically generating more training signals?” is Yes. As shown in Figure 2 and Table 5, ADPO produces more effective learning signals than DPO and consistently outperforms it on downstream tasks.
>
> > Weakness3: Provide standard deviation of the test performance.
>
> We now report standard deviations of all reported test accuracies.
> We observe that the variance is small and the relative ranking of methods remains unchanged, further supporting the robustness of our conclusions.
>
> |  | Llama-3-8B, GSM8K | Llama-3-8B, MATH | Gemma-3-12B, GSM8K | Gemma-3-12B, MATH | Qwen-3-8B, GSM8K | Qwen-3-8B, MATH | DS-Math-7B, GSM8K | DS-Math-7B, MATH |
> | --- | --- | --- | --- | --- | --- | --- | --- | --- |
> | DPO | 64.37±0.19 | 18.00±0.33 | 77.03±0.22 | 39.80±0.20 | 86.96±0.18 | 53.80±0.61 | 67.78±0.24 | 32.00±0.45 |
> | ADPO | 68.08±0.19 | 21.00±0.33 | 78.32±0.22 | 41.20±0.32 | 88.10±0.06 | 55.40±0.73 | 69.98±0.29 | 33.40±0.41 |
> | cDPO | 67.90±0.09 | 16.80±0.32 | 77.18±0.14 | 38.60±0.15 | 90.98±0.12 | 56.80±0.20 | 72.90±0.21 | 33.40±0.43 |
> | cADPO | 68.76±0.08 | 20.20±0.41 | 78.85±0.17 | 40.40±0.47 | 91.74±0.13 | 57.20±0.45 | 73.54±0.11 | 35.40±0.41 |
>
> |  | AlpacaEval2 LC | AlpacaEval2 WR | Arena-Hard WR | MT-Bench |
> | --- | --- | --- | --- | --- |
> | SFT | 26.0±0.75 | 25.3±1.28 | 22.3±1.5 | 6.9±0.0008 |
> | DPO  | 40.3±0.81 | 37.9±1.46 | 32.6±1.4 | 7.0±0.0008 |
> | SimPO  | 44.7±0.80 | 40.5±1.47 | 33.8±1.4 | 7.0±0.0008 |
> | ADPO | 45.8±0.79 | 41.1±1.49 | 34.4±1.4 | 7.1±0.0009 |
>
> > Question2: Do ADPO and DPO exhibit similar wall clock time for training? Does the training time of ADPO change along the change of granularity level?
>
> We report wall clock training times (seconds) for DPO and ADPO across different granularities.
> All methods exhibit similar training times (all differences are within 100 seconds), indicating that ADPO in any granularity level does not introduce significant computational overhead.
>
> |  | Llama-3-8B |
> | --- | --- |
> | ADPO (static, k=4) | 2314 |
> | ADPO (static, k=2) | 2228 |
> | ADPO (static, k=1) | 2301 |
> | ADPO (adaptive, m=1) (equivalent to DPO) | 2234 |
> | ADPO (adaptive, m=2) | 2287 |
> | ADPO (adaptive, m=16) | 2321 |
> | ADPO (adaptive, m=128) | 2318 |
> | ADPO (adaptive, m=256) | 2226 |
> | ADPO (adaptive, m=512) | 2327 |
>
> [1] "From r to Q: Your Language Model is Secretly a Q-Function", Rafailov et al., 2024.

---

> ### Author Response · Authors · 2025-11-25
> **Rebuttal by Authors (2)**
>
> > Question3: How does ADPO affect the overall response length?
>
> We measured the average output length across all models and datasets, and we consistently observed two trends: (a) ADPO produces slightly longer responses than DPO, while (b) cADPO produces shorter responses than cDPO. Below, we interpret these two observations:
>
> (a) Why ADPO > DPO in length?
>
> ADPO applies the preference comparison at each prefix, which encourages the model to accumulate positive evidence step-by-step rather than only at the sequence level. This finer-grained feedback naturally incentivizes the model to articulate intermediate reasoning more explicitly, resulting in slightly longer outputs.
>
> (b) Why cADPO < cDPO in length?
>
> cDPO identifies “critical tokens” (tokens that are strongly associated with correct / incorrect reasoning) and amplifies their penalties only at the level of the full sequence. This sequence-level penalty can unintentionally encourage the model to generate longer reasoning chains, because additional tokens provide more opportunities to “offset” or decrease the impact of critical-token penalties. In contrast, cADPO applies the same penalties locally at each prefix, so the model cannot mitigate them by increasing output length. This prefix-wise structure suppresses unnecessary elaboration and leads to more concise responses overall.
>
> |  | Llama-3-8B, GSM8K | Llama-3-8B, MATH | Gemma-3-12B, GSM8K | Gemma-3-12B, MATH | Qwen-3-8B, GSM8K | Qwen-3-8B, MATH | DS-Math-7B, GSM8K | DS-Math-7B, MATH |
> | --- | --- | --- | --- | --- | --- | --- | --- | --- |
> | DPO | 317.8 | 869.7 | 391.2 | 895.2 | 299.7 | 513.8 | 257.7 | 682.6 |
> | ADPO | 342.8 | 917.9 | 518.5 | 2192.3 | 294.9 | 525.7 | 280.8 | 662.9 |
> | cDPO | 520.4 | 1859.7 | 333.5 | 1174.9 | 299.8 | 526.9 | 261.7 | 696.7 |
> | cADPO | 308.1 | 847.0 | 287.2 | 841.0 | 313.8 | 510.4 | 250.3 | 630.0 |
>
> > Suggestion2: use the same color for the same in Figure2
>
> We updated the figure accordingly in the new manuscript.
>
> > Suggestion3: Present the performances of the base model, without any fine-tuning, in the experiment results
>
> We now provide the performance of the base model (before any fine-tuning) in Table 3.
> Note that for the conversation task (Table 5), “SFT” corresponds to the base model performance.
>
> |  | Llama-3-8B, GSM8K | Llama-3-8B, MATH | Gemma-3-12B, GSM8K | Gemma-3-12B, MATH | Qwen-3-8B, GSM8K | Qwen-3-8B, MATH | DS-Math-7B, GSM8K | DS-Math-7B, MATH |
> | --- | --- | --- | --- | --- | --- | --- | --- | --- |
> | Base | 56.40 | 16.80 | 73.46 | 41.00 | 86.28 | 53.40 | 64.10 | 31.40 |
> | DPO | 64.37 | 18.00 | 77.03 | 39.80 | 86.96 | 53.80 | 67.78 | 32.00 |
> | ADPO | 68.08 | 21.00 | 78.32 | 41.20 | 88.10 | 55.40 | 69.98 | 33.40 |
> | cDPO | 67.90 | 16.80 | 77.18 | 38.60 | 90.98 | 56.80 | 72.90 | 33.40 |
> | cADPO | 68.76 | 20.20 | 78.85 | 40.40 | 91.74 | 57.20 | 73.54 | 35.40 |

---

> > ### Comment · Reviewer_PGZo · 2025-11-26
> >
> > Dear authors,
> >
> > Thank you for your detailed response. I think the additional results well support the practical advantage of using ADPO over DPO and cDPO. I would like to request your further clarification on the following questions:
> >
> > 1. In a long response with many tokens, there can be a subsequence in the preferred response which actually the annotator did not prefer.  In addition, ADPO seems to assume that there are the same number of segments in the preferred and rejected responses, and compare them in the same order. Does ADPO guarantee that its segmentation always provide more accurate learning signals than DPO in these cases?
> >
> > 2. In practice, how do the users choose the right number of $k$ or $m$? Is the optimal value of segmentation not dependent on each prompt and provided responses?
> >
> > 3. It is nice that ADPO performs similarly to SimPO (overlapping test accuracy considering standard deviation), but it seems there are few different algorithms on the AlpacaEval2 leaderboard that used the same base model but had achieved better performances. Could you perhaps clarify how similar or different are the training settings of ADPO and SimPO models? How could the choice of ADPO be justified over other algorithms, when someone wants a preference optimization algorithm for their task?

---

> > > ### Author Response · Authors · 2025-11-27
> > > **Additional Rebutal by Authors (1)**
> > >
> > > Thank you very much for your detailed feedback.
> > > We appreciate the reviewer’s thoughtful comments, and we address each additional question below.
> > >
> > > # Response to Additional Question 1: Segmentation and Local Preference Inconsistencies
> > >
> > > > In a long response with many tokens, there can be a subsequence in the preferred response which actually the annotator did not prefer. In addition, ADPO seems to assume that there are the same number of segments in the preferred and rejected responses, and compare them in the same order. Does ADPO guarantee that its segmentation always provide more accurate learning signals than DPO in these cases?
> > >
> > > We would first like to clarify the reviewer’s concern.
> > >
> > > The potential issue does not arise simply from the fact that a long preferred response may contain a subsequence that the annotator would not actually prefer. Instead, the true source of the problem is the case where the preferred and rejected responses have substantially different lengths. In such cases, if segmentation is performed independently for each response, some segments may contain meaningful tokens in one response but only padding in the other, which can distort the margin comparison. We fully agree that such misalignment can weaken the learning signal if not handled properly.
> > >
> > > Below, we clarify that (1) the **Adaptive family fully resolves** this potential misalignment issue, and additionally, (2) ADPO yields **more robust prefix-level learning signals** than DPO in the scenario the reviewer described involving locally dispreferred subsequences within the chosen response.
> > >
> > > ## (1) Static vs. Adaptive segmentation: the concern does not arise in the Adaptive family
> > >
> > > In the **Static family,** segmentation is based on fixed-size token windows (k tokens).
> > > Thus, when the chosen and rejected responses differ in length, padding must be added to the shorter one, resulting in misaligned segments, for example, a padding-only segment paired with a meaningful segment (see the white vs. blue segments in Fig. 1(b)). In this case, the reviewer’s concern indeed applies.
> > >
> > > In contrast, the **Adaptive family** partitions both responses into the same number of segments, ensuring that segment boundaries remain aligned regardless of token length (Fig. 1(e–g)). Therefore, segmentation mismatch does not occur under this setting.
> > >
> > > Empirically, Tables 4 and 6 show a consistent trend across all models: DPO < ADPO (Static) < ADPO(Adaptive)
> > >
> > > This indicates that the Adaptive family not only avoids the misalignment issue but also provides the most effective learning signal in practice.
> > >
> > > ## (2) ADPO provides more robust signals than DPO for locally dispreferred subsequences
> > >
> > > > In a long response with many tokens, there can be a subsequence in the preferred response which actually the annotator did not prefer.
> > >
> > > The reviewer also raises the scenario where a preferred response contains locally undesirable prefixes. This case highlights another important difference between DPO and ADPO.
> > >
> > > DPO aggregates all token-level margins before applying the sigmoid (Eq. 6):
> > >
> > > $L_{\mathrm{DPO}} = -\log \sigma \left(\sum_i \Delta_i\right)$
> > >
> > > As a result, a negative margin at a “bad” prefix can be overpowered by large positive margins elsewhere, causing the gradient for that prefix to vanish once the global margin saturates the sigmoid. This makes DPO relatively insensitive to localized preference inconsistencies.
> > >
> > > In contrast, ADPO applies the sigmoid per-prefix (Eq. 17):
> > >
> > > $L_{\mathrm{ADPO}} = -\sum_i \log \sigma(\Delta_i)$
> > >
> > > Thus, each prefix contributes independently to the loss.
> > >
> > > Although the sigmoid is still monotonic and cannot eliminate all interactions, this structure substantially reduces the cancellation effect: negative margins cannot be fully overshadowed by positive margins in unrelated parts of the sequence.
> > > Consequently, ADPO yields more stable and prefix-sensitive gradients in exactly the type of scenario described by the reviewer.
> > >
> > > We do not claim a theoretical guarantee that ADPO always provides strictly more accurate signals, but its formulation is better conditioned to detect and penalize locally undesirable prefixes that would otherwise be washed out by DPO’s global aggregation.

---

> > > ### Author Response · Authors · 2025-11-27
> > > **Additional Rebutal by Authors (2)**
> > >
> > > # Response to Additional Question 2: Choosing the Segmentation Granularity (k or m)
> > >
> > > > In practice, how do users choose the right number of k or m?
> > >
> > > As shown in Section 6.2, the empirical trend is remarkably consistent across all models and tasks: finer granularity leads to better performance.
> > >
> > > - In the Static family, smaller k (e.g., k = 1 or 2) consistently improves over larger k.
> > > - In the Adaptive family, larger m (e.g., m = 128, 256, 512) consistently outperforms smaller m.
> > >
> > > This holds for both mathematical reasoning (Table 4) and conversational tasks (Table 6).
> > > Therefore, in practice, users can simply select small k (1–2) or large m (128–512), as these choices provide strong and robust performance without any per-prompt tuning.
> > > As noted in our response to Question 1, the Adaptive family might be better than Static family because it avoids segmentation misalignment when response lengths differ.
> > >
> > > > Is the optimal segmentation not dependent on each prompt and response?
> > >
> > > This is indeed an interesting research direction.
> > > The ADPO framework (Sections 5.2–5.3) allows the feedback-length measure μ′ to be defined arbitrarily, so segmentation could in principle depend on individual prompts or responses.
> > > For example, one could design algorithms that adaptively choose k or m, or even segment responses based on semantic units.
> > >
> > > However, we emphasize that such sophistication is not required to surpass DPO.
> > > Our current Static and Adaptive families, without any per-example adaptation, already produce substantial and consistent improvements over DPO across all benchmarks. Thus, exploring prompt-dependent or semantic-based segmentation is a promising future direction rather than a necessity for ADPO to be effective.
> > >
> > > #  Response to Additional Question 3: Settings of Conversational Task
> > >
> > > > Could you perhaps clarify how similar or different are the training settings of ADPO and SimPO models?
> > >
> > > For the conversation task, we followed the hyperparameter search procedure recommended in the official SimPO repository (https://github.com/princeton-nlp/SimPO).
> > > Specifically, we used β = 0.01 for all methods and searched learning rates in 3e-7, 5e-7, 8e-7, 1e-6.
> > > For SimPO, we additionally evaluated the configurations reported in the SimPO paper, including β = 2.5, β/γ = 0.55, and learning rate = 1e-6.
> > >
> > > Based on this search, we selected the following settings for the final comparison:
> > >
> > > - DPO: learning rate = 1e-6, β=0.01
> > > - ADPO: learning rate = 5e-7, β=0.01
> > > - SimPO: β = 2.5, β/γ = 0.55, learning rate = 1e-6 (reported values)
> > >
> > > All other aspects of training, including training data, number of epochs, optimizer configuration, and evaluation protocols, were kept identical across all methods.
> > > Thus, the comparison between ADPO and SimPO is strictly controlled and fair.
> > >
> > > > it seems there are few different algorithms on the AlpacaEval2 leaderboard that used the same base model but had achieved better performances
> > >
> > > We carefully checked the official AlpacaEval2 leaderboard (https://tatsu-lab.github.io/alpaca_eval/), but we could not find any entry that fine-tunes Llama-3-8B-Instruct using a preference-optimization under comparable settings (i.e., training on UltraFeedback with PairRM annotations).
> > >
> > > As far as we can determine, SimPO is the strongest publicly available DPO-variant applied to this base model. Therefore, our results, showing that ADPO achieves competitive or even superior performance compared to SimPO, demonstrate the effectiveness of ADPO under the same evaluation setting.
> > >
> > > > How could the choice of ADPO be justified over other algorithms, when someone wants a preference optimization algorithm for their task?
> > >
> > > To summarize, ADPO offers a theoretically grounded yet practical alternative to existing preference-optimization methods: it matches or surpasses DPO and SimPO under identical settings, while requiring no additional components or task-specific tuning. This makes ADPO a reasonable and appealing choice for practitioners who want a stable and general-purpose preference optimization algorithm.

---

> > > > ### Comment · Reviewer_PGZo · 2025-11-27
> > > >
> > > > Dear authors,
> > > >
> > > > Thanks again for your detailed response. I am really appreciating your effort in providing answers to my questions.
> > > > I will increase the score for this paper to 6 for now. I would appreciate the authors' further responses to the following questions and suggestions.
> > > >
> > > > ## Questions and Concerns
> > > >
> > > > 1. I still think adaptive segmentation can have issues as it **assumes that the preferred and rejected responses have the same number of segmentations.** As we do not specify any constrain in the length or the format of the provided responses, for the same prompt $x$, the candidate responses can be drastically different in terms of format and length. In this case, applying the adaptive segmentation does not guarantee the matched segmentations are comparable. While performing semantic segmentation for example will be a much trickier task, and empirical evaluations showed meaningful improvement from the original DPO, I think this concern is still valid.
> > > >
> > > > 2. I am maybe confused with how the concepts apply, but it seems the response the authors provided to my question regarding "dispreferred subsequences in the preferred response" is indicating something against ADPO. When the preference signals are only given at full response level, if locally "wrong" signals are preserved and potentially amplified compared to original DPO, is this not something undesirable?
> > > >
> > > > 3. It seems in the SimPO paper, they used responses generated from the base model with PairRM to form the preference dataset [1]. Did ADPO do the same? Or is the SimPO model you evaluated on AlpacaEval2 also trained with the original UltraFeedback dataset, without additional generation from the base model? If ADPO also relied on the base model generations, while discarding the original responses in the dataset, I think this should be mentioned in the paper.
> > > >
> > > > ## Suggestions
> > > >
> > > > 1. As the conversation task involves AlpacaEval2 evaluation, readers of the paper will like to see the proposed method outperforming more methods with the same base model. Testing ADPO in a similar setting to that of RainbowPO, obtaining preferences with ArmoRM instead of PairRM [2], might give us more exciting results.
> > > >
> > > > 2. It seems exceeding the page limit for main text is not allowed in the rebuttal period. It is currently over the limit by two lines and I recommend addressing this soon.
> > > >
> > > > [1] Meng, Yu, Mengzhou Xia, and Danqi Chen. "Simpo: Simple preference optimization with a reference-free reward." Advances in Neural Information Processing Systems 37 (2024): 124198-124235.
> > > >
> > > > [2] Zhao, Hanyang, et al. "Rainbowpo: A unified framework for combining improvements in preference optimization." arXiv preprint arXiv:2410.04203 (2024).

---

> > > > > ### Author Response · Authors · 2025-11-30
> > > > > **Official Comment by Authors**
> > > > >
> > > > > Dear Reviewer PGZo,
> > > > >
> > > > > Thank you very much for the positive update to your assessment and for providing additional helpful comments.
> > > > > Below, we address each question and suggestion.
> > > > >
> > > > > # Response to Question 1: On the Limitations of Adaptive Segmentation
> > > > >
> > > > > We fully agree with the reviewer’s concern.
> > > > > Adaptive segmentation guarantees that the preferred and rejected responses have the same number of segments, but it does not guarantee that the segments are semantically comparable when the two responses differ drastically in length, format, or structure. Such cases may indeed lead to imperfect alignment between segments.
> > > > >
> > > > > That said, in practical preference-optimization pipelines, including our experiments and prior DPO-based work[1], the chosen and rejected responses are typically generated by the same base LLM. As a result, large discrepancies in format or length between the two responses would be empirically rare. In line with this, and as the reviewer also acknowledges, our empirical evaluations consistently show that ADPO yields meaningful improvements over DPO despite this theoretical limitation.
> > > > >
> > > > > In summary, the reviewer’s concern is valid in principle, but in practical settings, we find that such severe mismatches occur infrequently, and ADPO remains effective and robust in real-world training scenarios.
> > > > >
> > > > > # Response to Question 2: On Dispreferred Subsequences in the Preferred Response
> > > > >
> > > > > We believe there may be a misunderstanding in our previous explanation.
> > > > > Our intention was not to suggest that ADPO “amplifies” locally dispreferred prefixes, but rather that ADPO preserves prefix-level preference signals that DPO necessarily collapses into a single global margin.
> > > > >
> > > > > This behavior is, in fact, desirable when the preferred response contains a locally suboptimal subsequence. For example, in such cases, a perfect preferred response and one containing a clear local defect should ideally be treated differently. Because DPO aggregates all token-level margins before the sigmoid, these two cases become relatively indistinguishable. ADPO instead allows the model to reflect differences within the response by retaining prefix-wise information, which aligns naturally with the autoregressive structure of the model.
> > > > >
> > > > > # Response to Question 3: Training Data Construction on Conversational Tasks
> > > > >
> > > > > Yes, in our experiments, we followed exactly the same data construction procedure as SimPO. Specifically, we generated multiple candidate responses from the base model and determined the chosen/rejected pairs using PairRM annotations. As suggested by the reviewer, we have clarified this in the revised manuscript to avoid any ambiguity.
> > > > >
> > > > > # Response to Suggestion 1:  Evaluation with ArmoRM Preferences
> > > > >
> > > > > We agree that constructing preference data with ArmoRM, as in RainbowPO, is a valuable direction and could provide an additional perspective on ADPO’s behavior.
> > > > > At the same time, our current experiments, conducted with the standard UltraFeedback + PairRM setup widely used in recent PO studies, already demonstrate that ADPO consistently improves over DPO and performs comparably to or better than SimPO on conversational benchmarks. We therefore believe that the effectiveness of ADPO in the conversational setting is sufficiently supported by our existing results.
> > > > >
> > > > > That said, we agree that evaluating ADPO with ArmoRM-based preferences would further strengthen the study. We plan to conduct these additional experiments and include the results in the camera-ready version or a future revision.
> > > > >
> > > > > # Response to Suggestion 2: Page Limit
> > > > >
> > > > > Thank you very much for the kind reminder regarding the page limit.
> > > > > However, according to the ICLR Author Guide (https://iclr.cc/Conferences/2026/AuthorGuide), the page limit is extended to 10 pages during the discussion/rebuttal phase. Thus, we believe that our current submission is within this updated limit.
> > > > >
> > > > >
> > > > > [1] Yu Meng, Mengzhou Xia, and Danqi Chen. SimPO: Simple Preference Optimization with a Reference-Free Reward. NerIPS2024.

---

### Comment · Area_Chair_JUNK · 2025-11-28
**Please Check the Authors' Responses**

Dear Reviewers,

The authors have posted their responses. Could you please take a moment to review their responses and check whether your concerns have been adequately addressed (if you have not done it yet)? If possible, kindly initiate the discussion at your earliest convenience.

Your timely assistance is essential for keeping the review process on track. Thank you very much for your support and contribution.

Best regards, Your AC

---

### Author Response · Authors · 2025-12-01
**Summary for the Area Chair**

Dear Area Chair,

We would like to thank the Area Chair for taking the time to consider our submission under the unusual circumstances of this year’s discussion process. As recommended by the organizers, we provide a brief summary of how our rebuttal and the reviewer discussions address all concerns raised in the initial reviews.

# Reviewer PGZo

Although the review score has been reverted, during the discussion period, we were able to address the reviewer’s concerns, and the reviewer explicitly updated the score from negative (4) to positive (6). The reviewer’s questions were fully answered in our responses.

# Reviewer mCzu

This reviewer already gave a positive initial score (6). We provided clarifications regarding the motivation and intuition behind ADPO, as requested, and no unresolved concerns remain.

# Reviewer waf7

This reviewer initially gave a negative score (4), but we thoroughly addressed all weaknesses and questions raised.

Specifically:

## 1. Implicit reward in ADPO (Weakness 1, Question 3)
- The reviewer asked for clarification and interpretation of the implicit reward in ADPO.
- We resolved this with a detailed clarification and an additional experiment illustrating prefix-wise reward behavior.

## 2. Characteristics of ADPO (Weakness 2, Question 2)
- The reviewer questioned the optimization behavior introduced by ADPO’s nonlinear transformation from DPO, particularly regarding reward behavior in specific cases and its effects on output length.
- We resolved this through a mathematical analysis highlighting ADPO’s advantages in these cases, as well as an additional experiment comparing the output lengths of DPO, ADPO, cDPO, and cADPO.

## 3. Comparison with token-level PPO (Weakness 3)
- The reviewer asked about the relationship between ADPO and token-level PPO and requested a comparison.
- We resolved this by clarifying that a direct comparison is out of scope and that reproducing token-level PPO is infeasible due to the lack of publicly available implementation details.

## 4. Segmentation and Adaptive family (Weakness 4, Question 4)
- The reviewer raised concerns about the segmentation strategy in ADPO and asked for possible new segmentation methods.
- We resolved this by clarifying how the Adaptive family directly addresses these concerns and by providing examples of alternative segmentation strategies.

## 5. Clarification of math problem formulation (Question 1)
- The reviewer asked for an experiment on math problems, noting that correctness/preference might be determined solely by the target prefix.
- We resolved this by clarifying that our math experiments were already conducted and that preference is not determined solely by the final answer; reasoning chains also influence output quality.

## 6. Computational cost (Question 5)
- The reviewer asked whether ADPO incurs additional computational cost.
- We resolved this with an additional experiment demonstrating that ADPO’s cost is comparable to that of DPO.

# Reviewer cRUn

This reviewer initially gave a negative score (2), but we addressed every concern in detail:

## 1. Clarification of theory (Weakness 1, Question 1)
- The reviewer requested clearer explanations of the theoretical construction.
- We resolved this through two detailed clarifications of the derivation and assumptions.

## 2. Conversational-task experiment setup (Weakness 2, 4; Question 2)
- The reviewer asked for clearer explanations of the training data, evaluation protocols, and reported values for conversational tasks.
- We resolved this with three clarifications describing the use of UltraFeedback as training data, the evaluation procedures, and the precise meaning of the reported metrics.

## 3. Additional baselines (Weakness 3)
- The reviewer asked for providing additional baselines: SimPO results for Table 3 and cDPO results for Table 5.
- We resolved this by running an additional experiment to add SimPO results and by clarifying that cDPO cannot be applied to conversational tasks due to its design.

## 4. Full-parameter training results (Question 4)
- The reviewer asked how ADPO behaves without LoRA.
- We resolved this by performing an additional full-parameter tuning experiment.

**Importantly, this reviewer explicitly wrote: “I am willing to change my opinion to accept if my concerns with regard to the experiment parts are well explained.”**

We believe that these experimental concerns have now been fully addressed through our responses in points 2, 3, and 4 above, and therefore **this reviewer’s score has substantial room for improvement under normal discussion conditions.**

In summary, we have carefully responded to every weakness and question raised by all reviewers and provided multiple additional experiments and clarifications where necessary. We hope that this summary assists the Area Chair in evaluating our submission together with the detailed rebuttal and discussion already present.

---

### Meta-Review · Area_Chair_g47U · 2026-01-09

**Summary:**

This paper was reviewed by four experts in the field. The recommendations are (2, 4, 4, 6). Most of the reviewers agree that the paper's quality is insufficient for publication and needs significant revision and careful polishing (experimental evaluation, paper presentation, comparison, etc.). For instance, Reviewer PGZo argues that citing and detailed comparisons with the original DPO paper are missing, and more experimental results and explanations are needed to support the claim. Reviewers waf7 and cRUn also share similar concerns that the paper presentation needs to be clear, and more experimental evaluations on additional results are needed. The manuscript needs to be further polished to make the motivation clearer and more convincing. Taking these concerns into consideration, the paper would not be accepted at this time. The authors are encouraged to consider the reviewers' comments when revising the paper for submission elsewhere.

**Reviewer Concerns:**

The authors have successfully addressed the concerns regarding missing references, citations, and the general presentation of the paper in their rebuttal. However, the experimental evaluation remains a critical bottleneck. Despite the revision, the additional comparisons and detailed evaluations provided are insufficient and not convincing. The empirical evidence is not yet convincing enough to fully validate the proposed method's superiority over existing baselines.

**Reviewer Scores:**

The reviewers would be satisfied with the author's commitment to adding missing references, discussions, and presentation in the revised version, while they may still hold concerns regarding the provided experimental evaluations and in-depth analysis.

---

### Decision · Program_Chairs · 2026-01-26

Reject